# ULTRA-360: Unconstrained Dataset for Large-scale Temporal 3D Reconstruction across Altitudes and Omnidirectional Views

**Xijun Liu**[1]*, **Zhaoliang Zhang**[1]*, **Yuxiang Guo**[1]*, **Yifan Zhou**[2], **Rama Chellappa**[1], **Cheng Peng**[3]

[1]Johns Hopkins University, Baltimore, MD, USA
[2]Zhejiang University, Hangzhou, China
[3]University of Virginia, Charlottesville, VA, USA

`{xliu253, zzhan288, yguo87, rchella4}@jhu.edu, yifanz@zju.edu.cn,`
`xuz7wn@virginia.edu`

## ABSTRACT

Significant progress has been made in photo-realistic scene reconstruction over recent years. Various disparate efforts have enabled capabilities such as multi-appearance or large-scale reconstruction from images acquired by consumer-grade cameras. How far away are we from digitally replicating the real world in 4D? So far, there appears to be a lack of well-designed dataset that can evaluate the holistic progress on large-scale scene reconstruction. We introduce a collection of imagery on a campus, acquired at different seasons, times of day, from multiple elevations, views, and at scale. To estimate many camera poses over such a large area and across elevations, we apply a semi-automated calibration pipeline to eliminate visual ambiguities and avoid excessive matching, then visually verify all calibration results to ensure accuracy. Finally, we benchmark various algorithms for automatic calibration and dense reconstruction on our dataset, named ULTRA-360, and demonstrate numerous potential areas to improve upon, e.g., balancing sensitivity and specificity in feature matching, densification and floaters in dense reconstruction, multi-appearance overfitting, etc. We believe ULTRA-360 can serve as the benchmark that reflect realistic challenges in an end-to-end scene-reconstruction pipeline.

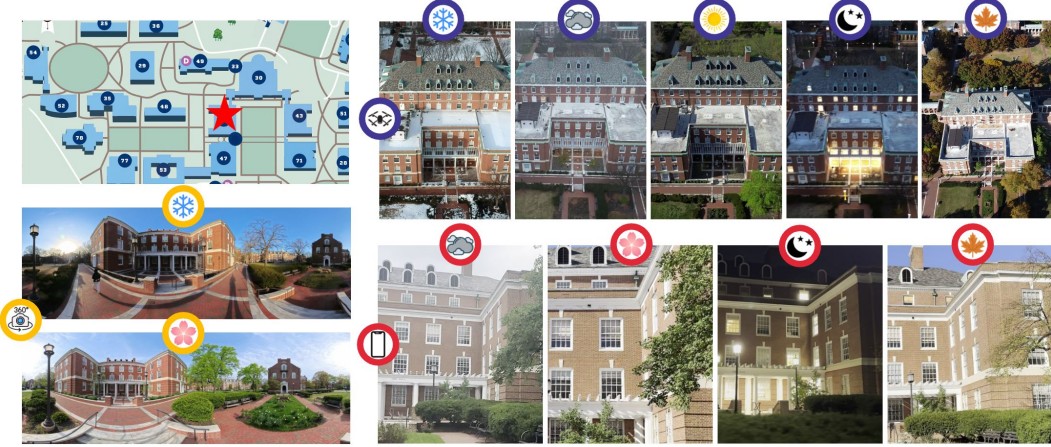

Figure 1: Sample images of one building, Building #10, in our imagery collected over multiple seasons, elevations, and multiple camera types to enable fully immersive 3D/4D reconstruction.

---

*Equal contribution. Xijun Liu: data acquisition, calibration pipeline development, and feed-forward registration experiments and evaluation. Zhaoliang Zhang: reconstruction experiments and evaluation. Yuxiang Guo: registration experiments and evaluation.

# 1 INTRODUCTION

Immersive digitalization of the 3D world is of great interests for Computer Vision and Graphics researchers, with many real-world applications in Robotics, AR/VR, Autonomous Driving, Urban Planning, etc. Tremendous progress has been made in recent years with neural rendering innovations. Methods such as Neural Radiance Field (NeRF) (Mildenhall et al., 2020) and 3D Gaussian Splatting (Kerbl et al., 2023) have improved quality in photorealistic scene reconstruction and novel-view synthesis. Various follow-up works have further extended this capability at larger scale (Turki et al., 2022; Liu et al., 2024a; Tancik et al., 2022b), to model sites across time (Martin-Brualla et al., 2021; Chen et al., 2022; Yang et al., 2023; Kulhanek et al., 2024b; Xu et al., 2024c; Zhang et al., 2024), and in various challenging scenarios, such as sparsity, incorrect exposure, or on degraded images (Zhu et al., 2023; Tang et al., 2025; Peng & Chellappa, 2023; Gao et al., 2024; Wu et al., 2025). With rapid progress and a vast amount of visual data online, we are closer than ever to achieving immersive 3D, perhaps even 4D, digitalization of the world. However, the current advances in neural reconstruction are often measured with disparate benchmarks and in specific aspects; collections of data from the internet (Snavely et al., 2006; Wallingford et al., 2024) are also difficult to be used for accurate evaluations, given the myriad of uncontrollable variables within the collection.

Pose and the Structure-from-Motion (SfM) point cloud from camera calibration are essential to dense reconstruction. Inverse rendering work typically assumes *known camera poses*, but how realistically can we assume accurate camera calibration, particularly for large scale scenes? Methods that address multi-view inconsistencies (Martin-Brualla et al., 2021; Chen et al., 2022; Yang et al., 2023; Kulhanek et al., 2024b; Xu et al., 2024c; Zhang et al., 2024) work well on small scale scenes with *dense* camera coverage. Are these methods adaptable to sparser coverage or in large scale scenes? While open source efforts (Tancik et al., 2023; Yu et al., 2022) have attempted to accommodate different methods and datasets, they have largely stalled as software complexity grows over time. How realistically can we reconstruct city-scale scenes with only 2D images, in an end-to-end manner? We summarize two limitations in the current benchmark datasets for calibration, dense reconstruction and Novel View Synthesis (NVS):

**Lack of Scale in Camera Coverage**. Current datasets are typically limited in two areas: the perspective camera format and the limited camera coverage. Perspective cameras are ubiquitous and easy to use; however, their limited Field-of-Views (FoV) lead to partial observation of the scene. As such, reconstructions are only viewable in one direction and are undesirable for immersive exploration. The distribution of cameras are also focused on one aspect of the scene, e.g. either on the ground (Tancik et al., 2022a; Meuleman et al., 2023) or in the air (Turki et al., 2022; Crandall et al., 2011). While aerial observations can recover large scale structures, ground observations contain much richer details. Additionally, the limited camera coverage makes NVS evaluation overly reliant on test cameras that are close to training cameras, and does not reveal issues such as obvious floaters (Warburg et al., 2023) in unconstrained exploration of the 3D asset.

**Lack of Scale in Realism and Time**. Synthetic data from unlimited perspectives and FoV can be generated from virtual engines (Li et al., 2023; Xiangli et al., 2022; Mittal et al., 2023), but such data lacks realism. Real images are full of inconsistencies that cannot be fully simulated, e.g., lighting, seasonality, weather, etc. So far, datasets that demonstrate these realistic scenarios (Snavely et al., 2006; Sabour et al., 2023) are small in scale and difficult to evaluate against. For example, Phototourism (Snavely et al., 2006) comprises of internet images collected at unknown time. As a result, methods (Martin-Brualla et al., 2021; Xu et al., 2024b; Kulhanek et al., 2024a; Xu et al., 2024a) developed on these datasets requires access to *test-view* images during evaluation to optimize appearance information. Various temporal concepts such as seasonality and structural modifications are neither fully transient nor based on only appearance changes.

We propose a dataset for Unconstrained Large-scale Temporal 3D Reconstruction across Altitudes, named *ULTRA-360*. ULTRA-360 is collected at a campus and aims to reconstruct and visualize the entire campus in 4D, with *hundreds of videos* collected across the span of *two years*, where the video frames are calibrated with *manual inspection* and aligned to a *consistent coordinate system*. ULTRA-360 provides:

1. **Immersive Ground Acquisition**: Both *perspective* and *360 panorama* images on the ground level are collected and calibrated to facilitate immersive 3D reconstruction.

2. **Multi-Elevation Acquisition**: Both *ground* and *aerial* images from multiple elevations are collected and calibrated to ensure full coverage of the buildings.

3. **Multi-Seasonality Acquisition**: Images are acquired across multiple seasons in a two year period, capturing the gradual changes over time.

4. **Large-Scale Calibration**: Twenty academic buildings are collected across multiple elevations, camera models, years, and are calibrated together.

We perform detailed evaluations of current State-of-The-Art methods on ULTRA-360, both in feature matching and dense reconstruction. The results reveal encouraging process and challenges to be addressed. For feature matching, recent innovations allow us to find correspondences across large distances. Despite such progress, SoTA feature matching methods (Lowe, 2004; DeTone et al., 2018; Edstedt et al., 2024b; Sun et al., 2021; Leroy et al., 2024; Sarlin et al., 2020) lie between the spectrum of *insufficient true positives* between images with large baselines and *significant false positives* between images with visual ambiguities. Scene graph optimization techniques (Cai et al., 2023; Xiangli et al., 2024; Arandjelovic et al., 2018; Duisterhof et al., 2024) can ameliorate such a process, but still requires various manual intervention.

Dense reconstruction from multiple elevations suffers from difficulties in sufficient densification and severe sky floaters, despite improvements to Level-of-Details. Multi-appearance modeling is often entangled with view-direction bias if treated as a per-image optimization. We make several modifications, including neural sky modeling and time-based appearance modeling, to tackle these issues, and expect future research to improve immersive reconstruction based on ULTRA-360.

## 2 RELATED WORK

Table 1: A comparison of existing multi-view datasets highlighting key properties, including scale, diversity of appearances, FoV, and viewpoint variation.

| Dataset | # images | Scale | Real/Synthetic | Time | Camera Type | Elevation |
|---|---|---|---|---|---|---|
| Phototourism (Snavely et al., 2006) | 150K | Scene | Real | Uncontrolled | Perspective | Ground |
| MegaScenes (Tung et al., 2024) | 2M | Scene | Real | Uncontrolled | Perspective | Ground |
| BlendedMVS (Yao et al., 2020) | 5K | Scene | Real+Synthetic | Single | Perspective | Ground |
| UrbanScene3D (Crandall et al., 2011) | 128K | Scene | Real+Synthetic | Single | Perspective | Aerial |
| Quad 6K (Crandall et al., 2011) | 5.1K | Scene | Real | Single | Perspective | Aerial |
| Mill 19 (Turki et al., 2022) | 3.6K | Scene | Real | Single | Perspective | Aerial |
| OMMO (Lu et al., 2023) | 14.7K | Scene | Real | Day/Night | Perspective | Aerial |
| Block-NeRF (Tancik et al., 2022b) | 2.8M | City | Real | Day/Night | 360 | Ground |
| KITTI-360 (Liao et al., 2023) | 300K | City | Real | Single | 360 | Ground |
| NuScenes (Caesar et al., 2020) | 1.4M | City | Real | Day/Night/Rainy | 360 | Ground |
| MatrixCity (Li et al., 2023) | 519K | City | Synthetic | Diff. Weather/Lighting | Perspective | Ground+Aerial |
| ULTRA-360 | 37.7 K | City | Real | Four Seasons, Day/Night | Perspective+360 | Ground+Aerial |

### 2.1 MULTI-VIEW DATASETS FOR DENSE RECONSTRUCTION

In scene reconstruction and NVS research, the widely used benchmark datasets often focus on single objects (Mildenhall et al., 2020; Knapitsch et al., 2017; Barron et al., 2022) or indoor scenes (Lin et al., 2018). These datasets are collected in controlled environments with accurate camera estimation. Various datasets (Snavely et al., 2006; Tung et al., 2024) construct outdoor unbounded architecture datasets with multi-view images from the internet. While these datasets include appearance diversity, the uncontrolled collection method leads to lack of multi-view imagery on a single consistent appearance. Consequently, algorithms (Martin-Brualla et al., 2021; Chen et al., 2022; Yang et al., 2023; Kulhanek et al., 2024b; Xu et al., 2024c; Zhang et al., 2024) tested on these datasets require access to test-view images during evaluation to account for unique appearance variation.

Large-scale datasets, such as Quad 6K (Crandall et al., 2011), UrbanScene3D (Crandall et al., 2011), Mill-19 (Turki et al., 2022), and OMMO (Lu et al., 2023), have been collected from an aerial platform. This limits the level of details in reconstructed models, if rendering or exploration from the ground perspective is desired. Driving datasets like Block-NeRF (Tancik et al., 2022b), KITTI-360 (Liao et al., 2023), and NuScenes (Caesar et al., 2020) focus on street-level imagery, leading to many unobserved regions such as the roof of the buildings. So far, no dataset has been proposed for a large-scale collection of imagery spanning multiple elevations. MatrixCity (Li et al., 2023) contains both ground and aerial imagery, but is synthesized through game engines.

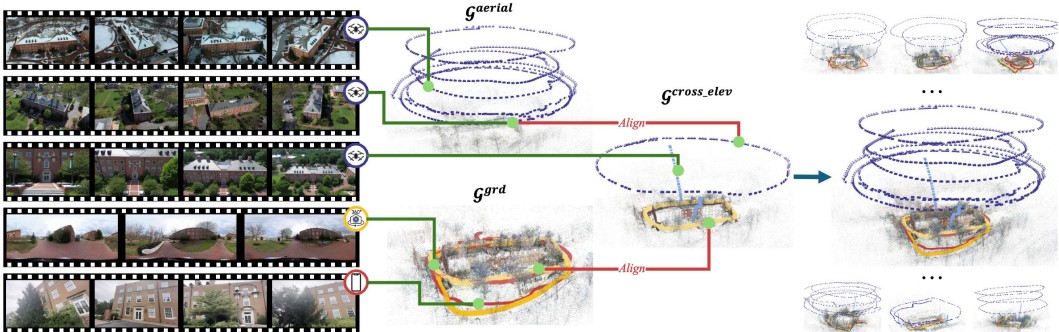

Figure 2: Visualization of the large-scale camera calibration process.

## 2.2 CAMERA CALIBRATION AND DENSE RECONSTRUCTION ALGORITHMS

Recovering dense 3D geometry from 2D images has a long history of research. Broadly speaking, camera calibration is first performed based on SfM (Schönberger & Frahm, 2016; Schönberger et al., 2016; Pan et al., 2024), which relies on reliable feature extractors (DeTone et al., 2018; Lowe, 1999; Edstedt et al., 2024a; 2025) and feature matchers (Sarlin et al., 2020; Lindenberger et al., 2023b; Sun et al., 2021; Edstedt et al., 2024b; Leroy et al., 2024) to find correspondences. SfM then performs triangulation to recover camera pose and sparse geometry. Such a calibration process can be computationally expensive or get stuck in incorrect solutions due to visual ambiguities; various scene graph optimization techniques (Arandjelovic et al., 2018; Berton et al., 2023; Cai et al., 2023; Xiangli et al., 2024) have been introduced to remove unnecessarily or ambiguous image pairs based on prior knowledge. Selecting the proper scene graph or feature matching algorithms are highly subjective and unpredictable; while datasets from the Image Matching Challenging (Bellavia et al., 2025) exist, they are not constructed to be also used for dense reconstruction evaluation.

Dense reconstruction, or photorealistic NVS, has progressed significantly with the introduction of NeRF (Mildenhall et al., 2020) and 3DGS (Kerbl et al., 2023). By optimizing an implicit or explicit radiance field on multi-view images through differential rendering, these methods can achieve photorealistic rednering quality. Follow-up works has improved upon NeRF (Turki et al., 2022; Tancik et al., 2022a; Mi & Xu, 2023; Reiser et al., 2023; Xiangli et al., 2022; Meuleman et al., 2023) and 3DGS (Lin et al., 2024; Liu et al., 2024b; Lu et al., 2024; Ren et al., 2024) in large scale reconstruction, e.g., by splitting the scene into multiple blocks for optimization, introducing Level-of-Detail rendering, multi-appearance modeling, etc. Evaluation is done on test cameras, typically in-between training cameras; however, no quantitative evaluation has been done on more free-formed and realistic novel views. As shown in Table 1, ULTRA-360 provides rich variations in both appearances, rendering FoVs, and cameras ranging from the ground to the sky, providing an unique opportunities to understand the effects of view-dependent effects, floaters, and details.

## 3 ULTRA-360

As shown in Figure 2, ULTRA-360 captures real-world, large-scale imagery with multi-appearance, multi-elevation, panorama coverage, and providing a comprehensive testing ground for evaluating modern scene reconstruction and NVS algorithms. This dataset contains over *30k calibrated images* on twenty academic halls within a campus, covering an area of approximately *140 acre* and a time period of *two years*. ULTRA-360 covers a variety of texture and material, e.g., grass, glass/windows, trees, rocks, etc., that are on the campus. In the following section, we describe the data collection process and the semi-automated calibration pipeline to construct ULTRA-360.

### 3.1 LARGE-SCALE DATA COLLECTION ACROSS TIME AND ELEVATION IN 360 DEGREES
Constructing a dataset for large scale, immersive 3D reconstruction over time is laborious, time-consuming, and computationally intensive. While professional photogrammetry software and devices exist, they are not scalable and difficult to integrate with novel research. To enable collection at scale in coverage and time, we elect to use a variety of consumer-grade devices and develop our own processing pipeline. As shown in Table 2, for each of the twenty buildings, we systematically collect both aerial and ground-level imagery across four seasons with different lighting conditions.

Table 2: Summary of ULTRA-360, where multi-view sequences are collected at different time, appearances, elevations, and FoVs.

| Device | # Videos | # Frames | Season | Appearance | FoV | Elevation |
|--------|----------|----------|--------|------------|-----|-----------|
| iPhone | 19 | 7134 | Summer, Fall | Sunny, Cloudy, Night | $70°$ | 0m |
| Insta360 | 31 | 23260 | Spring, Winter | Sunny, Cloudy, Night | $360°$ | 0m |
| DJI Mini 3 | 81 | 7334 | Spring, Winter | Sunny, Cloudy, Night | $82°$ | 60, 100, 120m |

For **ground** imagery, the data collection process involves walking around each building's perimeter with an iPhone or Insta360 camera to capture video sequences. We perform manual inspection on all extracted frames to remove low-quality images and ensure sufficient overlap. Particularly, panorama frames are split into four perspective images, each with a $120°$ FoV. These four frames together cover the horizontal $360°$ FoV around the camera. We discard the bottom face, which has a static human operator, and the top face, which is mostly sky. Any image that contains Personally Identifiable Information (PII), e.g., faces or vehicle license plates, are blurred through automated algorithms (Wu et al., 2019).

For **aerial** imagery, we operate DJI drones that follow a circular flight trajectory around the building. Drone flights are planned to ensure uniform and complete coverage. Multiple elevations are collected at 60, 100, and 120m. We also keep the ascending video sequences as the drone moves from ground level to approximately 60m above ground on two sides of each building. These ascending videos help improve calibration between ground and aerial imagery. From these videos, we sample individual frames, applying the same quality control measures as for ground-level data.

### 3.2 Semi-Automated Camera Calibration for Doppelganger Mitigation

After video acquisition and frame extraction, we build a semi-automated pipeline to obtain correct camera calibration for all images. Given the sheer size in the number of images and covered area, directly applying software, e.g., COLMAP (Schönberger & Frahm, 2016), is both infeasible and will lead to inaccurate results. As shown in Figure 2, we use a divide-and-conquer approach by 1. calibrating images within an elevation, 2. merging images across multiple elevations based on a manually verified cross-elevation set, and 3. merging images from different buildings into a single coordinate system.

**Image Calibration within an Single Elevation**. For camera calibration, a collection of images $\mathcal{I}$ are collected at different times. Based on these images, scene graphs $\mathcal{G}^{\text{grd}}$ and $\mathcal{G}^{\text{aerial}}$ can be constructed from the ground and aerial images. Scene graphs $\mathcal{G} = (\mathcal{I}, \mathcal{P})$ consist of $\mathcal{I}$ as nodes, and image pairs $\mathcal{P} = \{(\mathcal{I}_i, \mathcal{I}_j)\}$ as edges. Correspondences between $(\mathcal{I}_i, \mathcal{I}_j)$ are extracted if edge $\mathcal{P}$ exists; such correspondences are then used for triangulation in SfM. I.e., $\mathcal{G}$ determines the visibility of $\mathcal{I}$ to other images. Various implementations can be used to determine scene graph edges. Exhaustive scene graphs are generally more accurate, but can also lead to more false matches.

Visual ambiguity, often referred to as doppelgänger (Cai et al., 2023) matches, occur to cameras that are far apart due to their similar patterns. These doppelgängers are particularly common for ground image collection of *buildings*. Aerial images suffer less from visual ambiguities, as they have a more global view of the building. For $\mathcal{G}^{\text{aerial}}$, we simply use exhaustive matching. For $\mathcal{G}^{\text{grd}}$, we use a mixture of sequential and exhaustive scene graph constructions to avoid doppelgängers.

Specifically, we denote multi-appearance ground images as $\mathcal{I}_i^x$, where $x$ denotes the video sequence and $i$ denote the frame within the sequence. Image pairs $\mathcal{P} = \{\mathcal{P}_{\text{within}}^x\} \cup \{\mathcal{P}_{\text{between}}^{x,y}\}$ can be fully separated into pairs that are within sequence $x$ and between any two sequences $\{x, y\}$. For $\mathcal{P}_{\text{within}}^x$, we use sequential matching, i.e. $\mathcal{P}_{\text{within}}^x = \{(\mathcal{I}_i^x, \mathcal{I}_j^x) | |i - j| \leq 10\}$, which prevents far-away frames to match. Such a spatial constraint is harder to determine for $\mathcal{P}_{\text{between}}^{x,y}$, as different sequences may not follow the same path or pace. To this end, we manually bucket frames into $\mathcal{S}_{\text{front}}^x$ and $\mathcal{S}_{\text{back}}^x$, which denote frames that are looking at the *front* or *backside* of the building. $\mathcal{P}_{\text{between}}^{x,y}$ can then be define as:

$$\mathcal{P}_{\text{between}}^{x,y} = \{(\mathcal{I}_i^x, \mathcal{I}_j^y) | i \in \mathcal{S}_{\text{front}}^x, j \in \mathcal{S}_{\text{front}}^y\} \cup \{(\mathcal{I}_i^x, \mathcal{I}_j^y) | i \in \mathcal{S}_{\text{back}}^x, j \in \mathcal{S}_{\text{back}}^y\}. \quad (1)$$

We find this setup effectively eliminates cross-sequence doppelgangers, as visual ambiguity within the same side of the building can be controlled by spatial constraints of individual sequences. For

panorama images, which are split into four perspective frames, $\mathcal{P}_{\text{between}}^{x,y}$ against iPhone frames only involve the building-facing side of the panorama image.

**Cross-Elevation Calibration**. To connect calibrations from different elevations, we perform an additional calibration on a cross-elevation set. Specifically, we calibrate a panorama ground sequence with an aerial sequence. Registering cameras across a large baseline is challenging, due to a lack of sufficient correspondences. To assist cross-elevation calibration, we record two transitional drone sequences from ground to air for each building. Similar to ground images, transitional drone images can experience visual ambiguities at ground level. The two sequences are distributed on the front and backside of the building. We manually define the scene graph $\mathcal{G}^{\text{cross\_elev}}$, i.e.,

$$\mathcal{P}^{\text{cross\_elev}} = \{\mathcal{P}_{\text{grd}}^{\text{grd}}\} \cup \{\mathcal{P}_{\text{trans}}^{\text{grd}}\} \cup \{\mathcal{P}_{\text{aerial}}^{\text{grd}}\} \cup \{\mathcal{P}_{\text{trans}}^{\text{trans}}\} \cup \{\mathcal{P}_{\text{aerial}}^{\text{trans}}\} \cup \{\mathcal{P}_{\text{aerial}}^{\text{aerial}}\}, \tag{2}$$

where $\mathcal{P}_y^x$ denotes image pairs between two elevations (note that $\mathcal{P}_y^x \equiv \mathcal{P}_x^y$). For ground images, we apply sequential matching similar to the ground-only scenario previously, i.e., $\mathcal{P}_{\text{grd}}^{\text{grd}} \equiv \mathcal{P}_{\text{within}}^x$. We do not match ground and aerial images directly, i.e., $\mathcal{P}_{\text{aerial}}^{\text{grd}} = \emptyset$, as few accurate matches can be found and removing these pairs accelerate the feature matching process. Both $\mathcal{P}_{\text{aerial}}^{\text{trans}}$ and $\mathcal{P}_{\text{aerial}}^{\text{aerial}}$ are exhaustive. Finally, ground-transition and transition-transition pairings can be defined as:

$$\mathcal{P}_{\text{trans}}^{\text{grd}} = \{(\mathcal{I}_i^{\text{grd}}, \mathcal{I}_j^{\text{trans}}) | i \in \mathcal{S}_{\text{front}}^{\text{grd}}, j \in \mathcal{S}_{\text{front}}^{\text{trans}}\} \cup \{(\mathcal{I}_i^{\text{grd}}, \mathcal{I}_j^{\text{trans}}) | i \in \mathcal{S}_{\text{back}}^{\text{grd}}, j \in \mathcal{S}_{\text{back}}^{\text{trans}}\},$$
$$\mathcal{P}_{\text{trans}}^{\text{trans}} = \{(\mathcal{I}_i^{\text{trans}}, \mathcal{I}_j^{\text{trans}}) | i \in \mathcal{S}_{\text{front}}^{\text{trans}}, j \in \mathcal{S}_{\text{front}}^{\text{trans}}\} \cup \{(\mathcal{I}_i^{\text{trans}}, \mathcal{I}_j^{\text{trans}}) | i \in \mathcal{S}_{\text{back}}^{\text{trans}}, j \in \mathcal{S}_{\text{back}}^{\text{trans}}\}. \tag{3}$$

We use both SP+SG (DeTone et al., 2018; Sarlin et al., 2020) and RoMa (Edstedt et al., 2024b) to compute correspondences based on $\mathcal{P}^{\text{cross\_elev}}$, and COLMAP (Schönberger & Frahm, 2016) to perform SfM. Finally, we select the best results from different matchers.

## 3.3 COORDINATE ALIGNMENT

Since SfM systems estimate camera up to an *arbitrary* scale and orientation, we need to align multiple coordinate systems together. Given the same 3D points in two coordinate systems, Procrustes Alignment (Gower, 1975) finds the scale, rotation, and translation $\{s, r, t\}$ transformations between them:

$$s^*, r^*, t^* = \arg\min_{s,r,t} \sum_i \|s(rp_{\mathcal{X}}^i + t) - p_{\mathcal{Y}}^i\|^2, \tag{4}$$

where $p^{i,\mathcal{X}}$ and $p^{i,\mathcal{Y}}$ are 3D points in coordinate system $\mathcal{X}$ and $\mathcal{Y}$. To better align the camera systems, we optimize based on both the camera center $P_{\text{pos}}^{i,\mathcal{X}} \in \mathbb{R}^{1 \times 3}$ and rotation $R^{i,\mathcal{X}} \in \mathbb{R}^{3 \times 3}$. Specifically, we represent camera rotation by backprojecting three points based on camera center and rotation:

$$P_{\text{rot}}^{i,\mathcal{X}} = P_{\text{pos}}^{i,\mathcal{X}} + s^{\mathcal{X}} R^{i,\mathcal{X}}, P_{\text{rot}}^{i,\mathcal{Y}} = P_{\text{pos}}^{i,\mathcal{Y}} + s^{\mathcal{Y}} R^{i,\mathcal{Y}} \tag{5}$$

where $s^{\mathcal{X}} = \|\sigma^{\mathcal{X}}\|$, and $\sigma^{\mathcal{X}}$ is the standard deviations of $P_{\text{pos}}^{i,\mathcal{X}}$; $s^{\mathcal{X}}$ is similarly defined. Based on $P_{\text{rot}}^{i,\mathcal{X}}$ and $P_{\text{rot}}^{i,\mathcal{Y}}$, we update Eq. (4) as follows:

$$(s^*, r^*, t^*) = \arg\min_{s,r,t} \sum_i \|s(rP_{\text{pos}}^{i,\mathcal{X}} + t) - P_{\text{pos}}^{i,\mathcal{Y}}\|^2 + \|s(rP_{\text{rot}}^{i,\mathcal{X}} + t) - P_{\text{rot}}^{i,\mathcal{Y}}\|^2. \tag{6}$$

We show that this significantly improves the rotation alignment accuracy in our appendix.

**Single Building Alignment**. For every building, we obtain $(s_{\text{grd}}^*, r_{\text{grd}}^*, t_{\text{grd}}^*)$ from the ground-only coordinate system to the cross-elevation coordinate system. This is done by applying Eq. (6) on the panorama sequence, which are calibrated in both systems. Similarly, we find $(s_{\text{aerial}}^*, r_{\text{aerial}}^*, t_{\text{aerial}}^*)$ for aerial cameras based on the shared aerial sequence. All images of a single building can then be transformed into a unified coordinate frame.

**Campus-wide Alignment** To put cameras from all buildings into the same system, we perform a similar alignment process. To accomplish this, we first calibrate a subset of aerial images from every building, captured during summer from an altitude of 60m. Based on the shared aerial images, we use Eq. (6) to find the transformation of every building's individual coordinate system to the campus-wide aerial calibration.

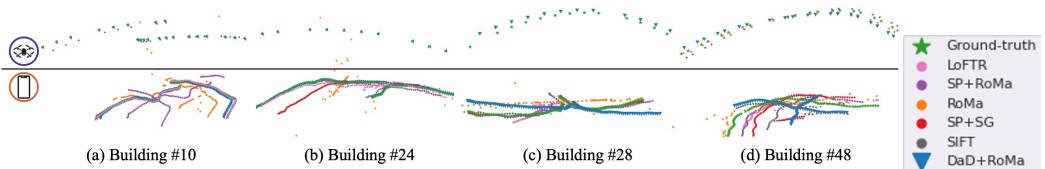

(a) Building #10    (b) Building #24    (c) Building #28    (d) Building #48

Figure 3: Visualization of multi-elevation camera poses obtained from different matching methods.

# 4 EXPERIMENTS

We examine SoTA camera calibration and dense reconstruction algorithms on ULTRA-360. Specifically, two challenges are explored in camera calibration: 1. finding true positive matches between far-apart images, e.g., across elevation; 2. avoiding false positive matches between images that are not visible to each other but have similar patterns. Two challenges are explored in dense reconstruction: 1. cross-elevation NVS and 2. multi-appearance NVS. Through experiments, we observe progress in these four challenges and many areas for future research to improve upon.

**Cross-Elevation Feature Matching**. For each building in ULTRA-360, we select a portion of the front side perspective ground images and aerial images acquired at 120m, *without* the transitional images that connect them. We test six popular feature matching algorithms: SIFT (Lowe, 2004), SP+SG (DeTone et al., 2018; Sarlin et al., 2020),SP+LG (DeTone et al., 2018; Lindenberger et al., 2023a), LoFTR (Sun et al., 2021), RoMa (Edstedt et al., 2024b), and RoMa filtered by two feature extractors, SuperPoint (DeTone et al., 2018) and DaD (Edstedt et al., 2025). Exhaustive matching is used for all scenarios mentioned above. In addition, we test four contemporary feed-forward matching methods: VGGSfM (Wang et al., 2024), VGGT (Wang et al., 2025), MASt3R (Leroy et al., 2024) and MASt3R-SfM (Duisterhof et al., 2025). We report AUC@10, computed from Relative Rotation Accuracy (RRA) and Relative Translation Accuracy (RTA). To isolate the cross-elevation challenge, AUC is computed only over ground–aerial pairs. For each ground–aerial pair, we measure the angular errors in rotation and translation and take the AUC of the minimum of RRA and RTA over 10-degrees threshold, a common metric for calibration.

As shown in Table. 3 and visualized in Fig. 3, calibrating cross-elevation images is challenging. In general, no algorithms can correctly calibrate all scenarios correctly. Interestingly, RoMa (Edstedt et al., 2024b)-based methods are the only ones with the ability to find cross-elevation correspondences. This can be attributed to its DINOv2 (Oquab et al., 2023) foundation model backbone. Despite the high sensitivity, RoMa (Edstedt et al., 2024b) is prone to false positives, as ground images are often falsely matched to each other due to similar patterns on the building. To this end, we find that SP (DeTone et al., 2018) or DaD (Edstedt et al., 2025) can help filter these false positives. However, they can still fail in Fig. 3(c) and (e).

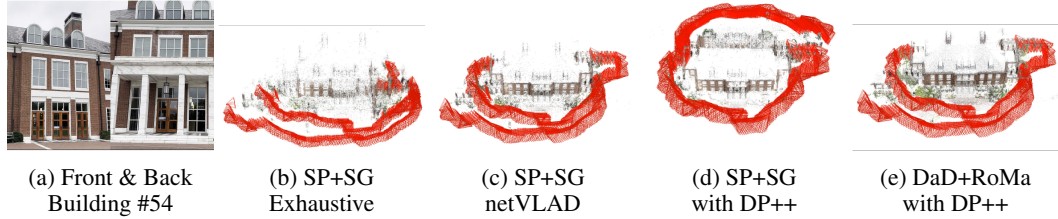

(a) Front & Back    (b) SP+SG    (c) SP+SG    (d) SP+SG    (e) DaD+RoMa
Building #54    Exhaustive    netVLAD    with DP++    with DP++

Figure 4: Visualization of calibration with various scene graph optimization methods given visual ambiguity in (a). All but (d) lead to suboptimal calibration solutions.

**Automated Scene Graph Optimization**. Various methods have been proposed to optimize the viewing scene graph to remove visually ambiguous pairs (Cai et al., 2023; Xiangli et al., 2024) and reduce excessive computation (Arandjelovic et al., 2018; Berton et al., 2023). These approaches are important for unconstrained calibration, where sensitive feature matchers are necessary, and false positive matches pose significant challenges. To this end, we evaluate several methods on ULTRA-360, particularly the ground panorama sequences.

As visualized in Fig. 4, exhaustive matching often leads to the worst results both in accuracy and computation due to visual ambiguities. NetVLAD (Arandjelovic et al., 2018) reduces computation

Table 3: Cross-Elevation camera poses obtained from different matching methods. Measured in AUC@10 (higher is better).

| Method | Building #10 | #24 | #28 | #34 | #48 | #49 | #54 |
|---|---|---|---|---|---|---|---|
| LoFTR | 0 | 0 | 0 | 0 | 0 | 0 | 0 |
| SP+RoMa | 0.3738 | 0 | 0 | 0 | 0.6986 | 0 | 0.5966 |
| RoMa | 0.0854 | 0.0023 | 0 | 0.0036 | 0.5030 | 0 | 0.1388 |
| SP+SG | 0 | 0 | 0 | 0 | 0 | 0 | 0 |
| SP+LG | 0 | 0 | 0 | 0 | 0 | 0 | 0 |
| SIFT | 0 | 0 | 0 | 0 | 0 | 0 | 0 |
| DaD+RoMa | 0.6941 | 0.8000 | 0 | 0.7915 | 0.5465 | 0.7440 | 0.6380 |
| VGGT | 0.1384 | 0 | 0 | 0 | 0.0003 | 0 | 0 |
| VGGSfM | 0 | 0 | 0 | 0 | 0 | 0 | 0 |
| MASt3R | OOM | 0 | 0 | 0 | 0 | 0 | 0 |
| MASt3R-SfM | 0 | 0 | 0 | 0 | 0 | 0 | 0 |

Table 4: Quantitative evaluation on multi-elevation reconstruction. We split the training set into either ground-only (G), aerial-only (A), or ground-aerial combined (GA) imagery. The test views are also separated into ground-only (G) and aerial-only (A) subsets. Due to different collection conditions, we only evaluate DSIM in *cross-elevation rendering*.

| Train | Test | Block-MERF | | | Splatfacto-W | | | CityGS V2 | | | Scaffold-GS | | | Octree-GS | | | EVER | | |
|---|---|---|---|---|---|---|---|---|---|---|---|---|---|---|---|---|---|---|---|
| | | PSNR | SSIM | DSIM | PSNR | SSIM | DSIM | PSNR | SSIM | DSIM | PSNR | SSIM | DSIM | PSNR | SSIM | DSIM | PSNR | SSIM | DSIM |
| G | G | 21.020 | 0.609 | 0.118 | 21.925 | 0.657 | 0.166 | 20.702 | 0.655 | 0.168 | 21.551 | 0.658 | 0.122 | 21.360 | **0.667** | **0.109** | **21.971** | 0.641 | 0.146 |
| A | G | ----- | ----- | 0.588 | ----- | ----- | 0.639 | ----- | ----- | **0.522** | ----- | ----- | 0.595 | ----- | ----- | 0.608 | ----- | ----- | 0.619 |
| GA | G | 19.655 | 0.574 | 0.235 | **21.569** | 0.647 | 0.183 | 20.585 | 0.643 | 0.188 | 21.140 | 0.635 | 0.154 | 21.184 | **0.653** | **0.116** | 21.522 | 0.624 | 0.175 |
| A | A | 27.451 | 0.779 | 0.015 | 29.440 | 0.860 | 0.016 | 28.997 | 0.840 | 0.009 | **30.286** | **0.878** | 0.006 | 29.950 | 0.874 | **0.005** | 26.397 | 0.720 | 0.023 |
| G | A | ----- | ----- | 0.847 | ----- | ----- | **0.714** | ----- | ----- | 0.743 | ----- | ----- | 0.822 | ----- | ----- | 0.755 | ----- | ----- | 0.740 |
| GA | A | 13.453 | 0.106 | 0.407 | 23.206 | 0.669 | 0.042 | 20.129 | 0.598 | 0.173 | 26.135 | 0.748 | **0.022** | **26.488** | **0.759** | 0.024 | 23.433 | 0.644 | 0.039 |

by cutting down unnecessary pairs, but cannot resolve doppelgangers. Doppelganger++ (Xiangli et al., 2024) simplifies the scene graph and address doppelgangers to some degree; however, sensitive matchers like RoMa (Edstedt et al., 2024b) still finds enough false matches to lead to a deformed calibration, whereas SuperGlue (Sarlin et al., 2020) is less sensitive but more specific, achieving the correct solution. In summary, selecting appropriate scene graphs and feature matchers to obtain good calibration still requires manual inspection and expertise. For more complete metrics and visualizations regarding image registration, please refer to our appendix.

**Large-scale Dense Reconstruction and NVS**. We select ten buildings from ULTRA-360 to evaluate current progress in robust, large-scale 3D reconstruction. For each building, we split training data into three configurations: 1. ground images only, 2. aerial images only, 3. both ground and aerial images. For each configuration, we evaluate from held-out ground and aerial cameras separately.

For baselines, we choose six SoTA methods for evaluation: Splatfacto-W (Xu et al., 2024a), Block-MERF (Song et al., 2024), CityGaussianV2 (Liu et al., 2024b), Scaffold-GS (Lu et al., 2024), Octree-GS (Ren et al., 2024) and EVER (Mai et al., 2024). Multiple metrics are used to evaluate NVS performance: Peak Signal-to-Noise Ratio (PSNR) and Structural Similarity (SSIM) are used for low-level quality evaluation. Perceptual similarity metrics DreamSim (Fu et al., 2023) (DSIM) are used to quantify semantic similarity, which helps in cases where pixel-wise groundtruth is not available due to e.g., changed lighting conditions.

As shown in Table 4, and visualized in Figures 5 and 6, we observe that Octree-GS (Ren et al., 2024) performs the best out of all methods, particularly when training data contain multi-elevation images. This can be attributed to its Level-of-Detail implementation. Scaffold-GS (Lu et al., 2024) achieves comparable fidelity through hierarchical Gaussian decomposition. All methods perform much worse given cross-elevation images for training compared to using single-elevation only. Interestingly, this may not be due to limited capacity. As shown in Table 5, cross-elevation reconstructions have significantly less Gaussians compared to single-elevation reconstructions, despite being strictly a superset in its training data. This likely indicates densification algorithms experience challenges when Gaussians' positional gradients are pulled from different directions.

By rendering cross-elevation cameras, we can observe various artifacts from current methods. Significant floaters exist when ground-only reconstructions are rendered from aerial perspectives. No-

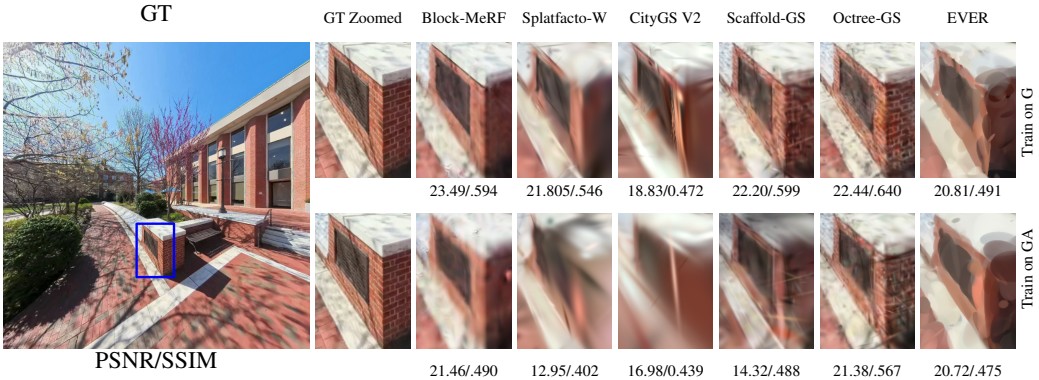

Figure 5: Visualization of ground image rendering from different reconstruction methods and two training configurations: ground-only images (G) and ground+aerial images (GA).

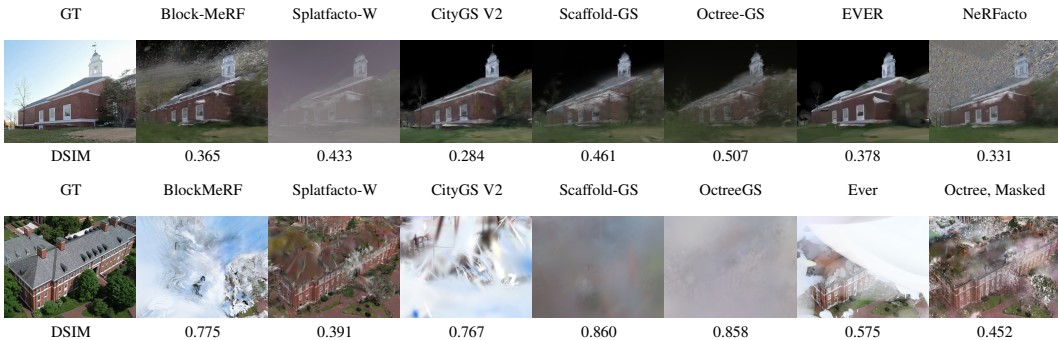

Figure 6: Top: ground view renderings from aerial-image-only reconstructions. Bottom: aerial view renderings from ground-image-only reconstructions.

tably, Splatfacto-W (Xu et al., 2024a) achieves superior aerial rendering through its background modeling. As shown in Figure 6, we also implement an implicit neural network to model sky in Octree-GS, which significantly reduce floaters. CityGS (Liu et al., 2024a;b) performs best on ground-view reconstruction from aerial data via geospatial-aware Gaussian priors optimized for large-scale aerial image.

**Multi-Appearance Reconstruction and Zero-Shot NVS**. ULTRA-360 contains multi-view sequences collected at different time. We use these sequences to evaluate multi-appearance reconstruction. Wild-GS (Xu et al., 2024b) and Gaussian-Wild (Zhang et al., 2024) are used as baselines, both of which require test image for evaluation. Unlike previous datasets (Snavely et al., 2006), ULTRA-360 has access to multi-view groundtruth at *every appearance*. This allows us to evaluate the effect of per-training-image embeddings on test images. As shown in in Table 6, we find that previous approaches lead to severe entanglement between view direction and the general appearances. Specifically, if we apply embedding from a training image that is the farthest away from the current test view, a significant drop in performance can be observed. The larger the performance drop suggests that the embeddings and networks are learned to overfit the input images, instead of the general 3D appearance. By modifying the per-image embedding to a time-based embedding, we can both remove the reliance on test-images at render time and achieve more 3D consistent appearance modeling. We provide more details and visualizations in our appendix.

Table 5: The average number of 3D Gaussians under different training configurations

| Train | Splatfacto-W | CityGS V2 | Octree-GS | EVER |
|---|---|---|---|---|
| G | 340244 | 569325 | 3191058 | 535701 |
| A | 630093 | 287026 | 527991 | 70738 |
| GA | 309018 | 241688 | 2230053 | 262366 |

Table 6: Quantitative evaluation of multi-appearance reconstruction and rendering based on ULTRA-360.

| | Wild-GS | | | Gaussian-Wild | | |
|---|---|---|---|---|---|---|
| | PSNR | SSIM | DSIM | PSNR | SSIM | DSIM |
| Test Image Embedding | 28.133 | 0.864 | 0.015 | 26.528 | 0.767 | 0.020 |
| Nearest Train Image Embedding | 28.003 | 0.863 | 0.014 | 26.567 | 0.757 | 0.020 |
| Farthest Train Image Embedding | 22.506 | 0.770 | 0.061 | 25.621 | 0.757 | 0.023 |
| Time Embedding | 27.973 | 0.860 | 0.014 | 26.277 | 0.762 | 0.021 |

## 5 Discussion and Conclusion

In this work, we propose a dataset called ULTRA-360 for Unconstrained Large-scale Temporal 3D Reconstruction across Altitudes. ULTRA-360 contains 37.7k frames collected from hundreds of videos across the campus and includes academic buildings from multiple seasons, multiple elevations, and multiple camera types. To this end, we ensure cameras from different elevations can find correspondences based on ground-to-aerial transitional images. We also eliminate false matches through manually defined scene graphs.

Popular feature matching and scene graph optimization algorithms are evaluated to measure how imagery from ULTRA-360 can be calibrated without assistance. Some methods demonstrate significant improvement in finding difficult true positives, at the cost of more false positives. While proper filtering based on keypoint extraction can lead to less false positives, current camera calibration pipeline still fall into incorrect solutions due to visual ambiguities, even with scene graph optimization. This showcases the need for a potentially more global approach in addressing doppelgangers rather than relying pair-wise prediction.

We also evaluate various dense reconstruction methods on ULTRA-360. We find that current methods, even those designed for large scale reconstruction, perform much worse given cross-elevation images for training compared to using single-elevation only. This likely indicates limitations in densification algorithms at scale. Multi-appearance reconstruction is also benchmarked. Several methods require access to test-time images to model appearance. Based on ULTRA-360, we find that these methods tend to generate embeddings that are heavily over-fitted to specific viewpoint, leading to suboptimal results to other views of the same appearance.

ULTRA-360 provides many novel directions for research, including the study on out-of-distribution NVS, campus-scale immersive 4D reconstruction, and potentially serving as test-grounds for evaluating geometric plausibility for generative models. In the future, we will continue to expand on ULTRA-360 to include more buildings and temporal variations.

## 6 Acknowledgement

This research is based upon work supported by the Office of the Director of National Intelligence (ODNI), Intelligence Advanced Research Projects Activity (IARPA), via IARPA R&D Contract No. 140D0423C0076. The views and conclusions contained herein are those of the authors and should not be interpreted as necessarily representing the official policies or endorsements, either expressed or implied, of the ODNI, IARPA, or the U.S. Government. The U.S. Government is authorized to reproduce and distribute reprints for Governmental purposes notwithstanding any copyright annotation thereon. The authors also thank the support and feedbacks from Dr. Ravi Ramamoorthi, at UCSD.

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

## A  APPENDIX / SUPPLEMENTAL MATERIAL

### A.1  RECONSTRUCTED CAMPUS VISUALIZATION

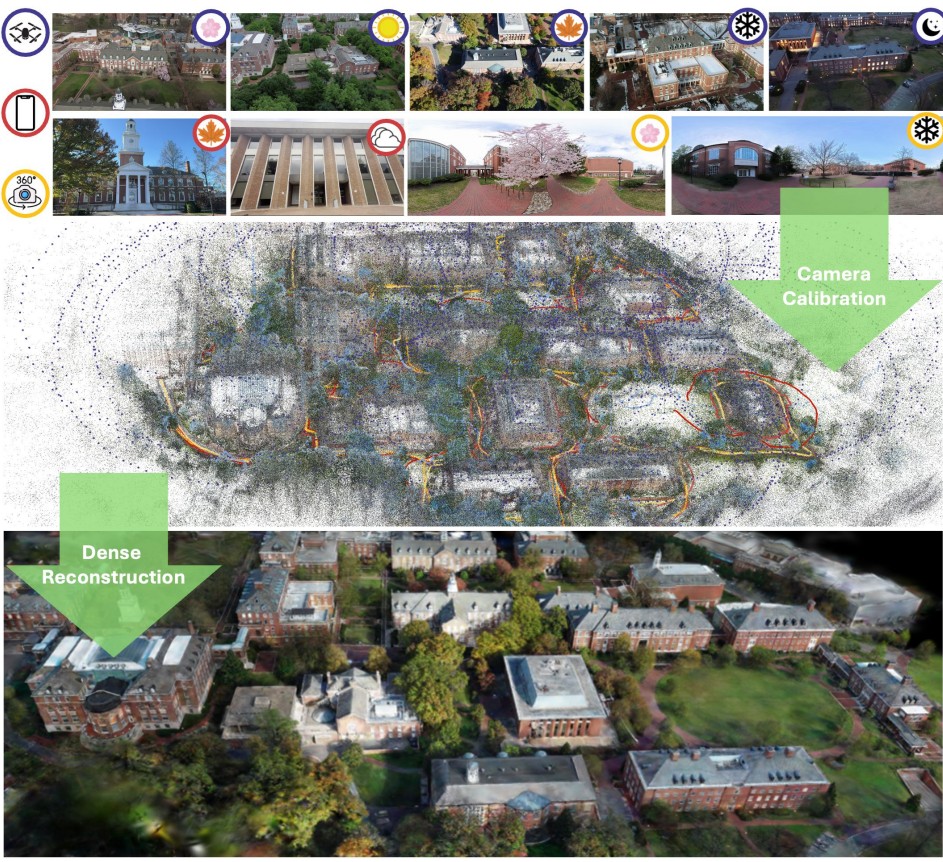

Figure A: A visualization of the reconstructed campus.

Figure A shows a visualization of the reconstructed campus based on our collected imagery over two years. The dataset is collected over multiple seasons, elevations, and multiple camera types to enable fully immersive 3D/4D reconstruction. All images have been calibrated into a unified coordinate system through a semi-automated process and manual verification.

### A.2  CROSS-ELEVATION FEATURE MATCHING

As shown in Figure B, we provide additional visualization of camera pose estimations for five buildings using six feature matching configurations, complementing the results shown in Figure 3. Overall, DaD+RoMa achieves higher accuracy, successfully estimating more camera poses with lower error. However, it fails to register the ground-level images in Figure B(a) and encounters false positive matches in Figure B(c), demonstrating the challenge of cross-elevation feature matching and underscoring the necessity of adopting the proposed single elevation calibration strategy.

We also visualize the absolute error of each estimated camera pose with respect to the ground truth after alignment in Figure C. Specifically, we sort these errors in ascending order; for images that fail to be calibrated, we assign a large error. DaD+RoMa is generally capable of estimating most camera poses except Figure C(d). Although SIFT struggles to register the multi-elevation images simultaneously, the successfully estimated poses tend to exhibit lower error, indicating higher confidence. We also observe that RoMa without any feature extractor leads to unstable results, which is reflected in the gradual increase in error across its estimated poses. This correlates with the observation that RoMa's raw correspondences contain both many true positives and false positives. In general, LoFTR and SP+SG performs similarly, compromising between sensitivity and specificity.

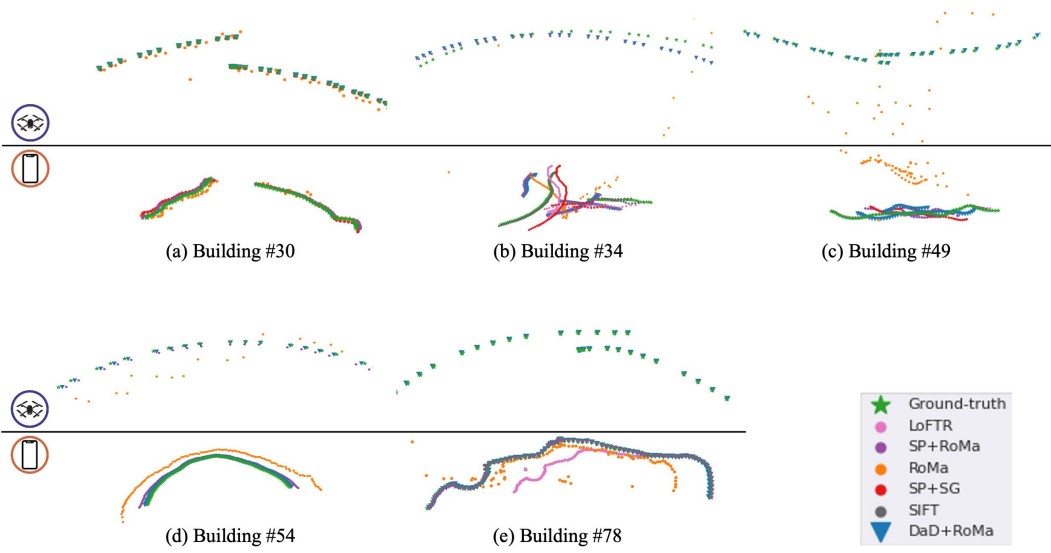

Figure B: Additional visualization of multi-elevation camera positions obtained from different matching methods.

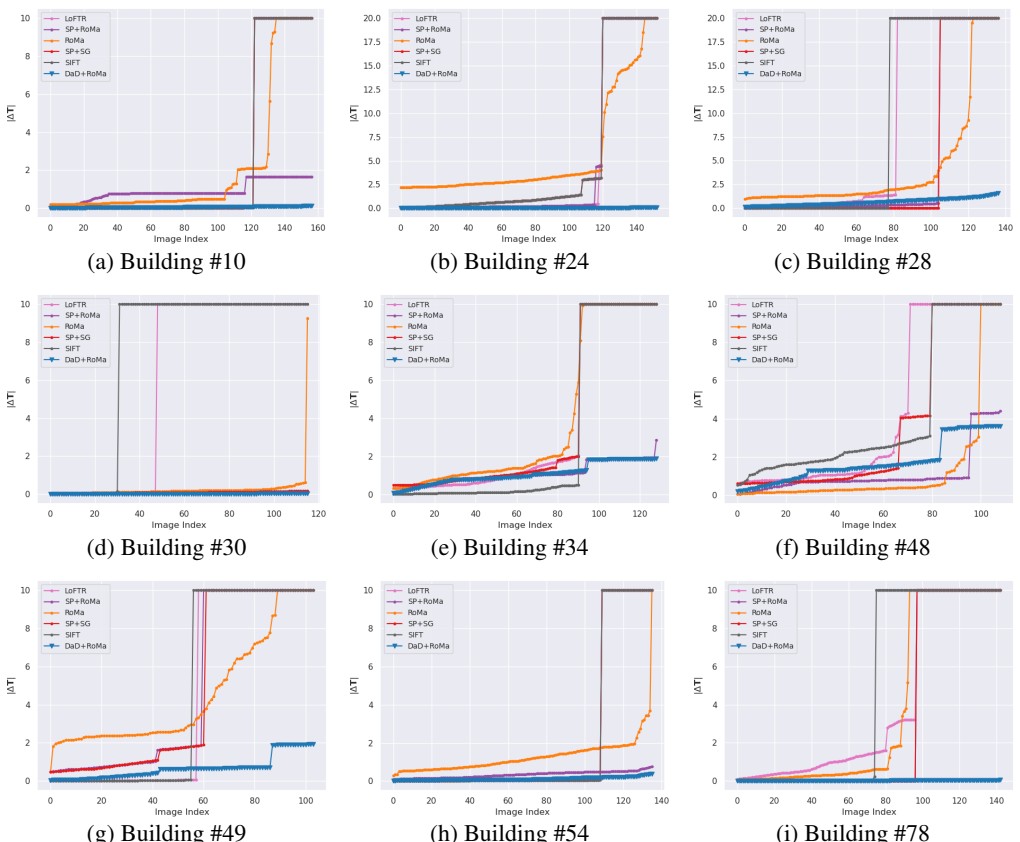

Figure C: Visualization of multi-elevation camera position error across nine buildings

## A.3 Automated Scene Graph Optimization

We provide more examples in Figure D to demonstrate the challenge in visual ambiguities. Many buildings look similar from different angles. Exhaustive matching, e.g., with SP+SG, often fails. Without any knowledge of acquisition time, netVLAD (Arandjelovic et al., 2018) sometimes can

help prune away unnecessarily pairs to achieve better reconstruction; however, it's also very unreliable. Doppelganger++ (Xiangli et al., 2024) does better at eliminating confusing pairs, but different feature matchers can still be prone to errors in different scenarios.

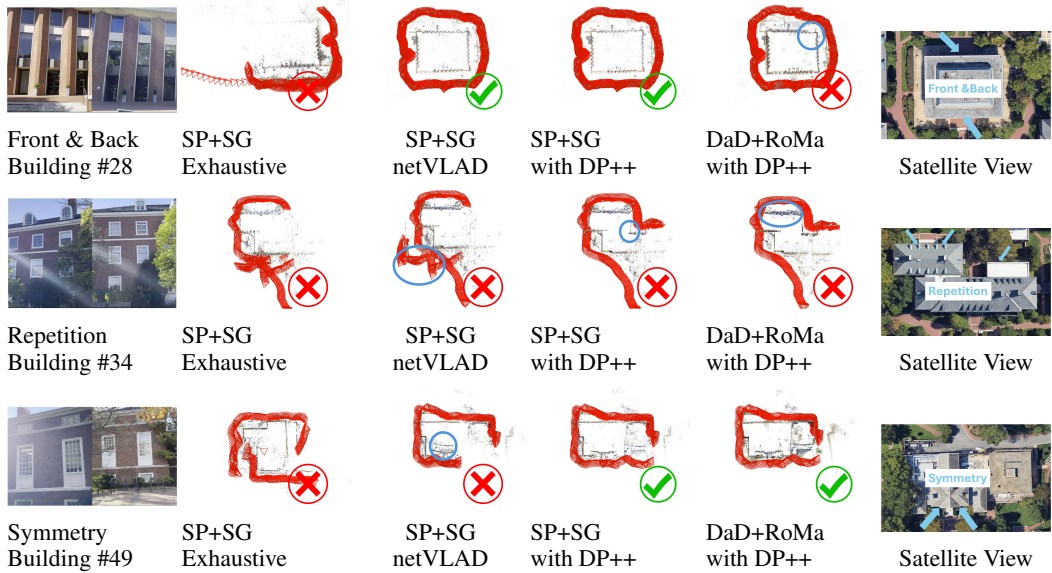

Figure D: Visualization of calibration with various scene graph optimization methods given visual ambiguity. A satellite view is provided to demonstrate the true locations of the images.

## A.4 LARGE-SCALE DENSE RECONSTRUCTION AND NVS

Our dense reconstruction evaluation uses PSNR, SSIM, and DSIM as primary metrics. As a perceptual similarity metric LPIPS (Zhang et al., 2018) is also included in Appendix Tables A and B for completeness.

Table A: LPIPS on multi-elevation reconstruction.

| Train | Test | Block-MERF | Splatfacto-W | CityGS V2 | Scaffold-GS | Octree-GS | EVER |
|-------|------|-----------|--------------|-----------|-------------|-----------|------|
| G | G | 0.513 | 0.522 | 0.512 | 0.483 | **0.443** | 0.467 |
| A | G | 0.899 | 0.846 | 0.861 | 0.871 | 0.881 | **0.829** |
| GA | G | 0.602 | 0.539 | 0.553 | 0.541 | **0.487** | 0.503 |
| A | A | 0.175 | 0.188 | 0.173 | **0.102** | 0.123 | 0.299 |
| G | A | 0.920 | 0.881 | 0.912 | 0.911 | **0.846** | 0.869 |
| GA | A | 0.708 | 0.394 | 0.532 | 0.277 | **0.266** | 0.355 |

Table B: LPIPS of multi-appearance reconstruction.

| | Wild-GS | Gaussian-Wild |
|---|---------|---------------|
| Test Image Embedding | 0.114 | 0.289 |
| Nearest Train Image Embedding | 0.115 | 0.288 |
| Farthest Train Image Embedding | 0.195 | 0.299 |
| Time Embedding | 0.118 | 0.298 |

## A.5 MULTI-APPEARANCE RECONSTRUCTION AND ZERO-SHOT NVS

Figure E shows the rendering results using different embeddings in the multi-appearance experiment. The difference maps between the rendered and ground truth images are also shown. It can be seen that the image rendered with the embedding farthest from the training view exhibits a significant overall appearance difference. These visual comparisons highlight a key drawback of per-image embeddings that they are view-dependent and lack consistency across different views.

## A.6 COORDINATE ALIGNMENT

| Mip-NeRF 360 | $E_R(\mu) \downarrow$ | $E_T(\mu) \downarrow$ |
|--------------|-----------------------|-----------------------|
| Procrustes Alignment | 0.196 | 0.0144 |
| + RANSAC | 0.179 | **0.0114** |
| *+ Rotation Points* | **0.156** | 0.0117 |

Table C: Improvements over Procrustes Alignment baseline in average rotation error $E_R$ and translation error $E_T$. Incorporating rotation points further minimizes the overall error.

We test the alignment algorithm on the Mip-NeRF 360 (Barron et al., 2022) dataset. Specifically, we calibrate a sparse subset of the images, then attempt to align it to the groundtruth coordinate

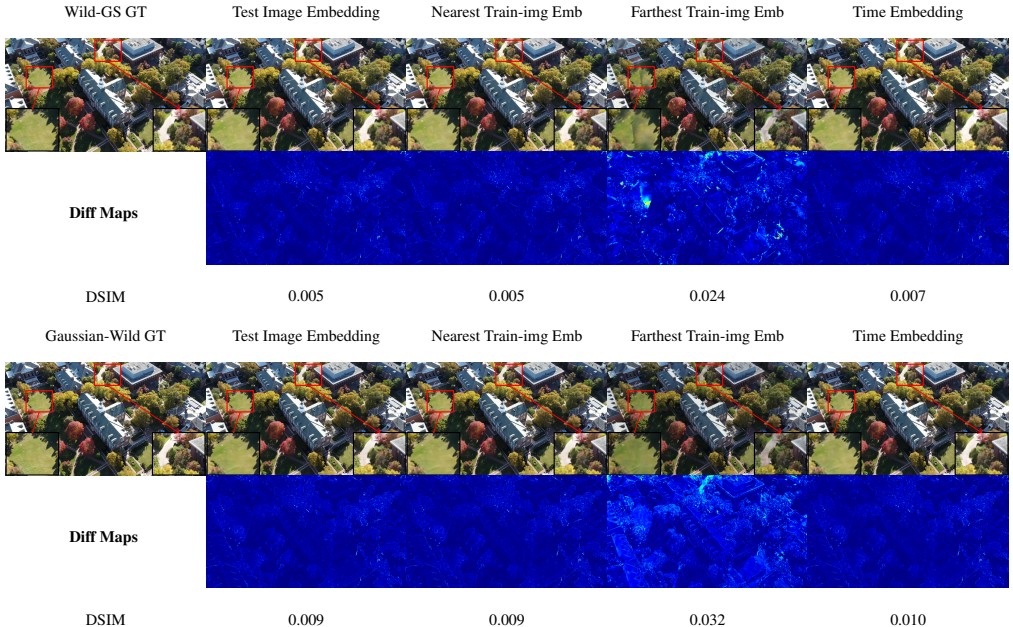

Figure E: The rendering result of Wild-GS and Gaussian-Wild on different appearance embeddings. Zoom-in images are shown in the bottom left and right; better viewed when magnified.

system. Sparse calibration leads to inaccuracy, and makes the alignment process more noisy. As shown in Tab. Table C, applying constraint on both the translation and rotation points indeed reduce the rotation error significantly.

## A.7 ADDITIONAL SCENE RECONSTRUCTION VISUALIZATION

Please refer to the videos for additional rendering of 3D structures of the campus buildings.

## A.8 THE USE OF LARGE LANGUAGE MODELS (LLMS)

We do not use any large language models in this work when constructing our dataset nor when drafting the paper.

