# OpenReview forum: "ULTRA-360: Unconstrained Dataset for Large-scale Temporal 3D Reconstruction across Altitudes and Omnidirectional Views"
_ICLR.cc/2026/Conference — ICLR 2026 Poster_

### Official Review · Reviewer_oFTk · 2025-10-25

**Soundness:** 3
**Presentation:** 3
**Contribution:** 3
**Rating:** 6
**Confidence:** 4

**Summary:**

The paper construct ULTRA-360, a video benchmark of buildings captured by different devices. The data of a building is captured in various seasons width different appearance from different elevations. The paper also presents a method to calibrate such a large number of images. Several methods are used as baselines to test on the ULTRA-360 for reconstruction.

**Strengths:**

The paper contributes a building images dataset, where every building is captured at different seasons, under various weather conditions, from different viewpoints. The data are curial for generating building appearance variations over time. These data are not rare.

**Weaknesses:**

Although the main advantage of such data is 4D evolution, the paper did not take 4D experiments. Only 3D reconstructions were evaluated.

Time evolutions of many kinds of data are important, but the dataset is only comprised of buildings. More complicated data are welcome, such as trees, whose appearance and geometric shape both change over time.

**Questions:**

Will these data be made publicly available? Will ground truth be available for buildings when the dataset is published?

What benefit is there in using images of buildings under different lighting conditions for 3D reconstruction? I think the images are useful for 4D reconstruction. However, I cannot find any benefits of these images for 3D reconstruction, except that they are troublesome.

I couldn't see obvious visual differences between the images in Figure E in the appendix.

---

> ### Author Response · Authors · 2025-11-21
> **Addressing questions from Reviewer oFTk**
>
> We thank Reviewer oFTk for his/her positive comments about our dataset's diversity. We will provide answers and clarifications below. We hope that they address the reviewer's concerns and would love to hear the reviewer's feedback.
>
> > Although the main advantage of such data is 4D evolution, the paper did not take 4D experiments. Only 3D reconstructions were evaluated. Time evolutions of many kinds of data are important, but the dataset is only comprised of buildings. More complicated data are welcome, such as trees, whose appearance and geometric shape both change over time.
>
> We performed explicit 4D reconstructions and introduced a temporal embedding to model 4D appearance changes. This is shown in Fig. E in supplemental material. Furthermore, all Ground+Aerial reconstruction experiments are also implicitly 4D, as appearance embeddings are needed to to address the different times in ground and aerial acquisition. E.g., as shown in Fig. 6 of the main manuscript, the reconstruction from ground (collected in winter) is visually different from the GT, which is taken from the air and in summer. This can be observed not only from buildings, but (precisely as the reviewer wants), from trees and vegetation.
>
> > Will these data be made publicly available? Will ground truth be available for buildings when the dataset is published?
>
> Yes! The dataset and groundtruth will be made publicly available after the review process.
>
> > What benefit is there in using images of buildings under different lighting conditions for 3D reconstruction? I think the images are useful for 4D reconstruction. However, I cannot find any benefits of these images for 3D reconstruction, except that they are troublesome.
>
>
> While we do focus on 4D reconstruction, our multi-elevation data provides a full coverage of the buildings and allows both the roof (not viewable from ground) and regions underneath the trees (not viewable from air) to be reconstructed in 3D. We note that multi-appearance capability is tightly coupled with multi-elevation reconstruction, as it is very challenging to obtain multi-elevation data at the exact same time/lighting condition.
>
> > I couldn't see obvious visual differences between the images in Figure E in the appendix.
>
> As we enlarged in the red box, you can see significant appearance differences in rendering from 1. the nearest train image appearance embedding and 2. the farthest train image appearance. Particularly, the correct appearance should be bright for the road, and uniform on the grass region, while the farthest image appearance rendering has a dimmer road and obvious visual artifacts on the grass.

---

> > ### Comment · Reviewer_oFTk · 2025-11-27
> >
> > I appreciate the authors' responses. I will maintain my score.

---

### Official Review · Reviewer_eWid · 2025-10-30

**Soundness:** 3
**Presentation:** 3
**Contribution:** 3
**Rating:** 6
**Confidence:** 3

**Summary:**

To address the limitations of datasets used in training NeRF and 3DGS, e.g., limited in scale, camera perspective diversity, the authors introduce ULTRA-360, a novel and comprehensive dataset designed to benchmark the holistic progress in large-scale, immersive 4D scene reconstruction. In addition, they conducted extensive comparisons using state-of-the-art methods on ULTRA-360 across two main tasks: camera calibration and dense reconstruction, even in some challenging settings. The observations from those experiments point the way for future research directions.

**Strengths:**

1. The motivation is strongly justified and clearly articulated. It effectively identifies specific, well-known gaps in existing datasets (lack of scale, elevation diversity, temporal realism) and connects these gaps to practical limitations in evaluating SOTA algorithms.
2. The ULTRA-360 dataset and its associated calibration pipeline is well-executed. The dataset's scale, multi-elevation, multi-season, and multi-camera nature is impressive. The semi-automated calibration pipeline, with its strategies for doppelganger mitigation and robust coordinate alignment, is a substantial technical contribution that adds great value to the dataset's utility and credibility.
3. The experimental section is comprehensive, evaluating a wide range of SOTA methods on clearly defined challenges (cross-elevation feature matching, scene graph optimization, large-scale dense reconstruction, multi-appearance reconstruction). The results are convincing and clearly demonstrate the shortcomings of current algorithms. The analysis is insightful, correctly attributing RoMa's performance to its DINOv2 backbone and identifying the overfitting problem in multi-appearance embeddings.

**Weaknesses:**

1. In experiments, some SOTA methods are not included, e.g., VGGT, Pi3.
2. The data is sourced solely from a single university campus environment, which limits scene diversity and the validation of model generalizability.
3. The heavy reliance on manual intervention (e.g., manually defining scene graphs) in the calibration pipeline contradicts the goal of fully automated and scalable data collection, potentially introducing subjectivity.
4. Its embodiment of "4D" is primarily confined to slow seasonal and illumination changes, lacking fast dynamics or geometric structural evolution in the scene.

**Questions:**

No more questions, see Weaknesses.

---

> ### Author Response · Authors · 2025-11-21
>
> We thank Reviewer eWid for his/her thoughtful comments. We are happy to see that the reviewer appreciates the clear motivation and strong, diverse benchmarking of the paper. The reviewer raised a series of questions, which we will provide answers and clarifications below. We hope that they address the reviewer's concerns and would love to hear the reviewer's feedback.
>
> 1. Experiments with VGGT and Pi3
> > In experiments, some SOTA methods are not included, e.g., VGGT, Pi3.
>
> We evaluate VGGT, Pi3 under the challenging scenarios, i.e. cross-elevation and doppelganger, revealed by ULTRA-360, with results reported in Table 1 and Table 2, respectively.
>
> In cross-elevation setting, the methods are tested on 5 buildings with 100+ images. For each building, we selected a subset of ground images and aerial images at 120m; these are the same settings used in our experiments, e.g., in Fig. 3 of main manuscript, Fig. B and C in Supplemental Material. We report AUC@10, computed from Relative Rotation Accuracy (RRA) and Relative Translation Accuracy (RTA). To isolate the cross-elevation challenge, AUC is computed only over ground–aerial pairs. For each ground–aerial pair, we measure the angular errors in rotation and translation and take the AUC of the minimum of RRA and RTA over 10-degrees threshold, a common metric for calibration.
>
> Under doppelganger regime, we reuse the setting from Fig. 4 of the main manuscript: Building #54, which has visually similar front and back doors and a total of 337 images.
>
> We did not include VGGT in the main manuscript primarily because of its large VRAM requirement: on our hardware, VGGT runs out of memory with only 100 images on a 48 GB GPU. In follow-up experiments on a H200 GPU, we evaluated both VGGT and Pi^3 on ULTRA-360. Pi3 is a concurrent work that came out close to our submission, but we include it here for completeness. Benefit from training on MatrixCity, which contains synthetic cross-elevation views, Pi3 performs noticeably better than VGGT on our cross-elevation benchmark. However, both VGGT and Pi3 fail catastrophically in the doppelganger setting. Moreover, Pi^3 remains significantly worse than RoMa based methods under cross-elevation scenario.
>
> Overall, these results support our claim that ULTRA-360 is a valuable benchmark that evaluates challenging failure modes for current feature matching and feedforward 3D methods.
>
> | Method     | B10 AUC@10 | B24 AUC@10 | B34 AUC@10 | B48 AUC@10 | B49 AUC@10 |
> |-----------|------------|------------|------------|------------|------------|
> | Pi3      | 0.2290     | 0.6274      | 0.1979          | 0.4078    | 0.3167          |
> | VGGT      | 0.1384     | 0            | 0          | 0.0003     | 0          |
> | VGGSfM   | 0          | 0               | 0          | 0          | 0          |
> | MASt3R    | OOM        | 0          | 0          | 0          | 0          |
> | MASt3R-SfM| 0          | 0           | 0          | 0          | 0          |
> | RoMA      | 0.0854     | 0.0023   | 0.0036     | 0.5030      | 0          |
> | SP+RoMa   | 0.3738     | 0         | 0          | 0.6986     | 0          |
> | DaD+RoMa  | 0.6941     | 0.8000    | 0.7915     | 0.5465     | 0.7440      |
>
> Table 1. Cross-Elevation camera poses obtained from different matching methods. Measured in AUC (higher is better).
>
> | Method        | AUC@30 | AUC@10 |
> |--------------|--------|--------|
> | MASt3R       | OOM    | OOM    |
> | MASt3R-SfM   | 0.2308 | 0.1706 |
> | VGGT         | 0.1936 | 0.1327 |
> | VGGSfM       | 0.1555 | 0.1291 |
> | Pi3  | 0.2175 | 0.1046|
>
> Table 2. Camera pose on Building #54, which reveals doppelganger regime, obtained from different matching methods. Measured in AUC (higher is better).

---

> ### Author Response · Authors · 2025-11-21
>
> 2. Diversity and generalizability
> > The data is sourced solely from a single university campus environment, which limits scene diversity and the validation of model generalizability.
>
> ULTRA-360 contains buildings of different shapes and sizes and captures a wide range textures including bricks, glass, vegetation, and roads etc. While the collection is concentrated in one geographic area, buildings in mostly every area must follow a consistent architectural pattern due to local regulations. Previous datasets cover iconic architectures or Metroplitan centers but do not reflect this reality well. In fact, finding feature correspondences is often much simpler in these datasets due to the unique structures, whereas in a regular setting, visual ambiguities (doppelgangers) can be found ubiquitously and pose significant challenges to reconstruction.
>
> Current reconstruction methods, e.g. NeRF and 3DGS, are optimization-based methods and are not generalizable to unseen regions. Supervised methods such as MAST3R and VGGT or depth estimation methods are generalizable; however, these methods are typically trained on a large swath of data. ULTRA-360 can serve as a valuable subset but should not be the only dataset. If combined properly, we do not see ULTRA-360 pose significant risks in reducing overall generalizability.
>
> 3. Manual intervention
> > The heavy reliance on manual intervention (e.g., manually defining scene graphs) in the calibration pipeline contradicts the goal of fully automated and scalable data collection, potentially introducing subjectivity.
>
> ULTRA360 consists of images that are difficult to match, i.e., scenarios such as cross-elevation matching, doppelganger scenes, and large-scale camera calibration. To the best of our knowledge, no existing method can calibrate ULTRA-360 end-to-end without manual intervention under these conditions. In order to achieve the best groundtruth for evaluation purpose, some manual intervention is required to ensure reliability and accuracy. The manual process is rather light, and in fact significantly reduces the workload for data collection. Without such process, significantly denser overlap and accurate GPS measurements will be required. We plan to publish all processing code and parameters in the final release, including multi-elevation alignment, such that future researchers can replicate our collection process.
>
> 4. Geometric structural evolution
> > Its embodiment of "4D" is primarily confined to slow seasonal and illumination changes, lacking fast dynamics or geometric structural evolution in the scene.
>
> We agree that our dataset mainly focuses on seasonal and appearance changes, we note that there are a rich set of geometric structures that do evolve over time. These structures include trees and vegetation, which undergo both appearance and geometry changes, temporary structures, e.g. tents built for certain events and removed later, and construction over time, particularly focused on two to three buildings. Since these attributes are often unexpected, temporary, and difficult to record systemically, we do not explicitly highlight them. However, we are happy to include a few examples and visualizations of such data if the reviewer finds them useful. It is true that our dataset does not record fast dynamics (e.g., rapid movements under 10 seconds); the temporal scale in our dataset is in months/seasons, which is a contrasting and unique feature compared to datasets of rapid movement.

---

### Official Review · Reviewer_o2rg · 2025-10-31

**Soundness:** 3
**Presentation:** 2
**Contribution:** 2
**Rating:** 4
**Confidence:** 4

**Summary:**

The authors present ULTRA-360, a campus-scale dataset that contains 37.7 k images extracted from hundreds of videos recorded over two years. Data span four seasons, three elevations (ground, 60 m, 100–120 m) and two device classes (perspective and 360° cameras). A semi-automatic COLMAP-based calibration pipeline is proposed to register all views into a single coordinate frame, followed by an empirical survey of six recent NeRF / 3D-Gaussian-splatting variants on cross-elevation and multi-appearance novel-view synthesis. The work is positioned as a benchmark that reveals “realistic challenges” in end-to-end 4D scene reconstruction.

**Strengths:**

1.	Scale & diversity: To date, this is the largest real-capture repository that combines ground-level 360° imagery, drone footage, dense time sampling, and seasonal variation for the same physical site.
2.	Thorough calibration effort: The authors manually verified every building-level reconstruction and performed cross-elevation alignment.

**Weaknesses:**

1.	Novelty is primarily dataset-oriented, with limited methodological innovation. While the paper introduces a valuable large-scale dataset, it does not propose new algorithms, loss functions, or theoretical frameworks.
2.	The calibration pipeline is an engineering assembly of existing components (COLMAP, SuperPoint/SuperGlue, RoMa) plus manual clean-up;
3.	Feature-matching ablations omit contemporary SOTA baselines such as MASt3R[1], DUSt3R[2].
4.	The observation that “cross-elevation training reduces Gaussians” is reported but not explained; no ablation of densification schedule, initialization, or gradient statistics is given.
[1] Grounding Image Matching in 3D with MASt3R
[2] DUSt3R: Geometric 3D Vision Made Easy

**Questions:**

1. What are the technical novelties in the paper?
2. Conduct comparative experiments with MASt3R and DUSt3R.

---

> ### Author Response · Authors · 2025-11-21
>
> We thank Reviewer o2rg for his/her thoughtful comments. We are happy to see that the reviewer believes this is a timely dataset paper with diverse imagery with rigorous calibration effort. This is a sentiment shared by all reviewers. The reviewer raised a series of questions, which we will provide answers and clarifications below. We hope that they address the reviewer's concerns and would love to hear the reviewer's feedback.
>
> 1. Technical novelty
> > What are the technical novelties in the paper?
>
> We note that our submission is under ICLR's Primary Area of **Datasets and Benchmarks**, hence our goal in this work is to introduce a novel dataset and benchmark that explore the lack of a standardized dataset for challenging scenarios for 3D vision. The main body of work explains the validity of the data collection pipeline (such that this process may be replicable to researchers in the future), and the problems revealed through our benchmarking.
>
> However, **there are technical improvements** in this paper outside of the dataset and benchmark content to demonstrate the utility of the ULTRA dataset. As shown in Fig. E in our Supplemental Material, we revealed the spatial-appearance inaccuracy in per-image appearance embedding, and introduced an improvement on 3DGS in-the-wild by leveraging a temporal embedding, which performs just as well as per-image appearance but generalize to different poses. This is only made possible by our dataset, which has multi-view images at different time stamp, unlike Photo Tourism.
>
> We also integrated a sky segmentation model to help with floaters in NVS. Pertinent to another question posed by the reviewer, ULTRA is a strong benchmark for evaluating floaters. Floaters are hard to pinpoint, as geometry is always unknown in inverse rendering (and impossible to obtain groundtruth at scale in such an unconstrained scenario). However, ULTRA datasets contain views that are far from each other, i.e., ground and aerial views. As such, floaters that arise from ground reconstruction can be seen easily from the air, as shown in Fig. 6 in the main manuscript. This is similar to Nerfbusters (2304.10532), which has a much smaller set of data with smaller scene scale.

---

> ### Author Response · Authors · 2025-11-21
>
> 2. Experiments with MASt3R and DUSt3R
> > Conduct comparative experiments with MASt3R and DUSt3R.
>
> While methods like DUST3R, MASt3R and VGGT are relevant to our work, our manuscript did not include these methods because 1. It is challenging to evaluate these methods due to their feedforward nature and large VRAM requirement, and 2. they all failed catastrophically, as shown in Table 1 and Table 2 below.
>
> Conventional SfM, e.g. COLMAP has a fixed GPU upper bound based on two-view matchers, while methods like DUST3R, MASt3R, VGGT have unbounded memory footprint with even a modest number of images. E.g., MASt3R 's global alignment will lead to OOM with just 50 images on a 24GB GPU, VGG-SfM and VGGT will lead to OOM with 100 images on a 48GB GPU. Given our large-scale calibration effort, e.g., with thousands of images at a time, these methods are simply infeasible.
>
> Perhaps more importantly, **none of these methods work** for the scenarios that ULTRA360 poses, i.e. cross-elevation and DoppelGanger, which was evident to us with limited testing. To demonstrate this point quantitatively, we managed to gain access to a H200 GPU and evaluated MASt3R, MASt3R-SfM, VGGT and VGG-SfM.
>
> We test the performance of these methods under **cross-elevation** scenario on 7 buildings with 100+ images. For each building, we selected a subset of ground images and aerial images at 120m; these are the same settings used in our experiments, e.g., in Fig. 3, Fig. B and C in Supplemental Material. We report AUC@10, computed from Relative Rotation Accuracy (RRA) and Relative Translation Accuracy (RTA). To isolate the cross-elevation challenge, AUC is computed *only over ground–aerial pairs*. For each ground–aerial pair, we measure the angular errors in rotation and translation and take the AUC of the minimum of RRA and RTA over 10-degrees threshold, a common metric for calibration. As shown in Table 1 below, MASt3R, MASt3R-SfM, VGGT, VGG-SfM, and SuperPoint+LightGlue all fail on almost all scenes. RoMa-based methods perform significantly better, though still not perfectly. We note that, while RoMa performs well here, it fails catastrophically in the presence of doppelgangers (Fig. 4 in the main manuscript, even after applying DP++)
>
> - Why does modern methods like MASt3R, MASt3R-SfM, VGGT not work in challenging settings?
>    - These feedforward 3D foundation models still have a domain gap to novel scenarios like cross-elevation images.
>    - These methods calibrate based on aligning depth with depth confidence; in cross-elevation or large-baseline scenarios, depth overlap is very small and depth confidence is low, leading to nothing to align with.
>
> We reuse the setting from Fig. 4 of the main manuscript: Building #54, which has visually similar front and back doors and a total of 337 images. We evaluate SuperPoint+LightGlue (SP+LG), MASt3R, MASt3R-SfM, VGGT, and VGGT-SfM on this scene to assess their robustness in the **doppelganger** regime, and report AUC@30 and AUC@10. As shown in Table 2 here, apart from MASt3R, which runs out of memory, most methods fail to reliably distinguish the front door from the back door. SP+SG with DP++ performs well here, but not perfectly, as shown in Fig. C in Supplemental Material. Furthermore, SP+SG fails in cross-elevation scenarios. Again, our dataset reveals the tradeoff between sensitivity and specificity in unconstrained, real world scenarios.
>
> We are happy to include Table 1 and Table 2 in the final manuscript, which support our claim that ULTRA-360 presents a unique dataset that demonstrates the complex trade-off between specificity and sensitivity in calibration methods.

---

> ### Author Response · Authors · 2025-11-21
>
> | Method     | B10 AUC@10 | B24 AUC@10 | B28 AUC@10 | B34 AUC@10 | B48 AUC@10 | B49 AUC@10 | B54 AUC@10 |
> |-----------|------------|------------|------------|------------|------------|------------|------------|
> | VGGT      | 0.1384     | 0          | 0          | 0          | 0.0003     | 0          | 0          |
> | VGGSfM   | 0          | 0          | 0          | 0          | 0          | 0          | 0          |
> | MASt3R    | OOM        | 0          | 0          | 0          | 0          | 0          | 0          |
> | MASt3R-SfM| 0          | 0          | 0          | 0          | 0          | 0          | 0          |
> | SP+LG     | 0          | 0          | 0          | 0          | 0          | 0          | 0          |
> | SIFT      | 0          | 0          | 0          | 0          | 0          | 0          | 0          |
> | LoFTR     | 0          | 0          | 0          | 0          | 0          | 0          | 0          |
> | SP+SG     | 0          | 0          | 0          | 0          | 0          | 0          | 0          |
> | RoMA      | 0.0854     | 0.0023     | 0          | 0.0036     | 0.5030      | 0          | 0.1388     |
> | SP+RoMa   | 0.3738     | 0          | 0          | 0          | 0.6986     | 0          | 0.5966     |
> | DaD+RoMa  | 0.6941     | 0.8000        | 0          | 0.7915     | 0.5465     | 0.7440      | 0.6380      |
>
> Table 1. Cross-Elevation camera poses obtained from different matching methods. Measured in AUC (higher is better).
>
>
> | Method        | AUC@30 | AUC@10 |
> |--------------|--------|--------|
> | SP+LG           | 0.2699 | 0.2593 |
> | MASt3R       | OOM    | OOM    |
> | MASt3R-SfM   | 0.2308 | 0.1706 |
> | VGGT         | 0.1936 | 0.1327 |
> | VGGSfM       | 0.1555 | 0.1291 |
> | DaD+RoMa with DP++  | 0.4897 | 0.476  |
> | SP+SG with DP++     | 0.9583 | 0.9341 |
> | SP+SG with NetVLAD | 0.4967 | 0.482  |
>
> Table 2. Camera pose on Building #54, which reveals doppelganger regime, obtained from different matching methods. Measured in AUC (higher is better).

---

> ### Author Response · Authors · 2025-11-26
>
> Dear Reviewer o2rg,
>
> We kindly invite you to review our rebuttal since the discussion phase is ending soon. We have carefully addressed all your concerns, including:
> 1.	Validity of the cross-view and doppelganger calibration challenges presented by this dataset, especially in relationship to contemporary feed-forward matchers.
> 2.	Novelty
>
> If you have any further questions or would like additional clarifications, we would be happy to continue the discussion.

---

### Official Review · Reviewer_S12u · 2025-11-04

**Soundness:** 3
**Presentation:** 3
**Contribution:** 2
**Rating:** 4
**Confidence:** 3

**Summary:**

The paper introduces ULTRA-360, a campus-scale dataset and benchmark targeting unconstrained, temporal, and multi-elevation 3D/4D reconstruction. The data spans ~140 acres over ~2 years, covering 20 academic buildings with both ground (perspective & 360°) and aerial (60/100/120 m) imagery acquired using consumer devices (iPhone, Insta360, DJI Mini), yielding >30k calibrated images (later summarized as 37.7k frames) with PII blurred. The authors build a semi-automated calibration pipeline: (i) within-elevation calibration via carefully constrained scene graphs to mitigate “doppelgänger” ambiguities, (ii) cross-elevation registration using transitional drone sequences from ground to air, and (iii) campus-wide coordinate alignment. They then benchmark feature matching (SIFT, SP+SG, LoFTR, RoMa; plus scene-graph optimizers such as NetVLAD and Doppelgänger++) and large-scale neural rendering methods (Block-MERF, Splatfacto-W, CityGaussianV2, Scaffold-GS, Octree-GS, EVER). Key findings: RoMa is the only method reliably finding cross-elevation correspondences but is prone to false positives; Octree-GS tends to perform best with mixed elevations; cross-elevation training often hurts densification; time-based appearance embeddings reduce overfitting relative to per-image embeddings.

**Strengths:**

Overall, this is a timely dataset paper with a clear gap and careful engineering. I appreciate it in the following aspests:

1.Clearly motivated and realistic benchmark for end-to-end large-scale reconstruction across elevations and seasons; combines perspective and 360° ground coverage with aerial views.

2.Thoughtful calibration strategy addressing doppelgänger matches via side-aware sequential pairing and cross-elevation transitional sequences; sensible divide-and-conquer pipeline.

3. Provide strong, diverse baselines & analysis covering both matching/scene-graph choices and modern 3DGS/NeRF variants, with concrete takeaways (e.g., RoMa sensitivity vs. specificity; Octree-GS advantages with multi-elevation).

4. A temporal appearance study with a simple but insightful time-based embedding alternative that reduces view-direction overfitting compared to per-image embeddings.

5. Privacy consideration via automated PII blurring in the released frames.

**Weaknesses:**

Although the topic is timely, the core contribution is largely data collection plus a semi-automated calibration pipeline with several heuristic/manual steps, which limits both novelty and technical depth for a ICLR main-track paper. The work would read stronger in a benchmarks/datasets track or at a vision/graphics venue focused on 3D reconstruction. Reproducibility/logistics also feel under-specified (clear license, canonical train/val/test splits by building/elevation/season, and standardized compute budgets/hyperparameters for baselines), which makes it hard to ensure fair future comparisons.

While realistic, ~37.7k frames over ~20 buildings is modest relative to recent city-scale datasets such as Mapillary Metropolis, StreetGaussians, DL3DV-10K, or 4D4NeRF; the paper offers little direct comparison or clear positioning against them. It also omits strong contemporary cross-view/cross-modality matchers (e.g., MASt3R/DUSt3R, LightGlue, VGGSfM/++), leaving the central “cross-elevation” claim under-tested. The authors should articulate where ULTRA-360 is uniquely diagnostic (e.g., specific failure modes it exposes) and define standardized tasks/splits (no-transition cross-elevation calibration; ground→aerial / aerial→ground NVS; seasonal generalization) with apples-to-apples baselines to justify its distinct value.

Here are some minor weakness:

1. Manual components remain central (manual bucketing of “front/back” sides; manual verification of cross-elevation sets), which could limit scalability/reproducibility and make results sensitive to human curation.

2. Calibration quality reporting is mostly qualitative; clearer quantitative pose/rotation accuracy and failure analysis (per building/elevation) would strengthen claims about robustness. (The paper notes remaining ambiguities even with scene-graph optimization.)
3. Cross-elevation dependency on transitional sequences may understate the difficulty when such data are unavailable; direct ground↔aerial matching is intentionally disabled as infeasible.
4. Evaluation scope: image-quality metrics (PSNR/SSIM/DSIM) are reported, but free-navigation artifacts (e.g., floaters) are not quantitatively assessed; geometric plausibility metrics are suggested as future work.

**Questions:**

1. Beyond visualizations, can you report quantitative pose errors (e.g., ATE/RE) against a reference (e.g., campus-wide aerial bundle) and per-building success rates for different scene-graph/matcher settings? This would help substantiate the semi-automated pipeline’s reliability.

2. How sensitive is performance to the front/back bucketing and the sequential window size |i−j|≤10? Could a learned side-classifier (or GPS/compass when available) reduce manual curation? Please provide failure cases and statistics.

3. Since you disable ground↔aerial matching (P_grd^aerial = ∅) as unreliable, what happens if only ground+high-altitude imagery exist (no transitional videos)? A “no-transition” benchmark variant and results would be very informative.

4. You already include SIFT, SP+SG, LoFTR, RoMa. Could you add/compare LightGlue/MASt3R-SfM and report robustness vs. false positives in ULTRA-360’s doppelgänger regimes? (You discuss scene-graph choices; numbers for these newer matchers would round this out.)

5. Consider a floater index (e.g., depth variance or 3D occupancy regularity under aerial renders of ground-only models) and a geometric plausibility score for free-navigation views. Even simple proxies would give a quantitative handle on the artifacts you highlight.

6. You note cross-elevation training reduces Gaussian counts and likely stresses densification. Can you isolate whether this is due to optimization schedules, regularization, or scene-graph/pose noise? An ablation on densification thresholds and background/sky modeling across methods would clarify attribution.

**Details Of Ethics Concerns:**

Campus imagery may include bystanders and vehicles. You blur PII (faces, plates), which is good; still, please clarify detector specifics, thresholds, manual QA, and policies for takedown requests. Also clarify whether minors might appear and how you handle that.

---

> ### Author Response · Authors · 2025-11-20
> **Addressing questions from Reviewer S12u**
>
> We thank Reviewer S12u for his/her thoughtful comments. We are happy to see that the reviewer believes this is a timely dataset paper with clear motivation and strong, diverse baselines/benchmarking. This is a sentiment shared by all reviewers. The reviewer raised a series of questions, which we will provide answers and clarifications below. We hope that they address the reviewer's concerns and would love to hear the reviewer's feedback.
>
> 1. **The validity of the cross-view calibration challenges presented by this dataset, especially in relationship to contemporary matchers**
> > the central “cross-elevation” claim under-tested, the authors should articulate where ULTRA-360 is uniquely diagnostic
>
> While methods like DUST3R, MASt3R and VGGT are relevant to our work, our manuscript did not include these methods because 1. It is challenging to evaluate these methods due to their feedforward nature and large VRAM requirement, and 2. they all failed catastrophically, as shown in Table 1 below.
>
> Conventional SfM, e.g. COLMAP has a fixed GPU upper bound based on two-view matchers, while methods like DUST3R, MASt3R, VGGT have unbounded memory footprint with even a modest number of images. E.g., MASt3R 's global alignment will lead to OOM with just 50 images on a 24GB GPU, VGG-SfM and VGGT will lead to OOM with 100 images on a 48GB GPU. Given our large-scale calibration effort, e.g., with thousands of images at a time, these methods are simply infeasible.
>
> Perhaps more importantly, **none of these methods work** for the scenarios that ULTRA360 poses, which was evident to us with limited testing. To demonstrate this point quantitatively, we managed to gain access to a H200 GPU and evaluated MASt3R, MASt3R-SfM, VGGT, VGG-SfM, and SuperPoint+LightGlue on 7 buildings with 100+ images.
>
> For each building, we selected a subset of ground images and aerial images at 120m; these are the same settings used in our experiments, e.g., in Fig. 3, Fig. B and C in Supplemental Material. We report AUC@10, computed from Relative Rotation Accuracy (RRA) and Relative Translation Accuracy (RTA). To isolate the cross-elevation challenge, AUC is computed *only over ground–aerial pairs*. For each ground–aerial pair, we measure the angular errors in rotation and translation and take the AUC of the minimum of RRA and RTA over 10-degrees threshold, a common metric for calibration. As shown in Table 1 below, MASt3R, MASt3R-SfM, VGGT, VGG-SfM, and SuperPoint+LightGlue all fail on almost all scenes. RoMa-based methods perform significantly better, though still not perfectly. We note that, while RoMa performs well here, it fails catastrophically in the presence of doppelgangers (Fig. 4 in the main manuscript, even after applying DP++)
>
> We are happy to include Table 1 in the final manuscript, which support our claim that ULTRA-360 presents a unique dataset that demonstrates the complex trade-off between specificity and sensitivity in calibration methods.
>
> - Why does modern methods like MASt3R, MASt3R-SfM, VGGT not work in challenging settings?
>    - These feedforward 3D foundation models still have a domain gap to novel scenarios like cross-elevation images.
>    - These methods calibrate based on aligning depth with depth confidence; in cross-elevation or large-baseline scenarios, depth overlap is very small and depth confidence is low, leading to nothing to align with.
>
> | Method     | B10 AUC@10 | B24 AUC@10 | B28 AUC@10 | B34 AUC@10 | B48 AUC@10 | B49 AUC@10 | B54 AUC@10 |
> |-----------|------------|------------|------------|------------|------------|------------|------------|
> | VGGT      | 0.1384     | 0          | 0          | 0          | 0.0003     | 0          | 0          |
> | VGGSfM   | 0          | 0          | 0          | 0          | 0          | 0          | 0          |
> | MASt3R    | OOM        | 0          | 0          | 0          | 0          | 0          | 0          |
> | MASt3R-SfM| 0          | 0          | 0          | 0          | 0          | 0          | 0          |
> | SP+LG     | 0          | 0          | 0          | 0          | 0          | 0          | 0          |
> | SIFT      | 0          | 0          | 0          | 0          | 0          | 0          | 0          |
> | LoFTR     | 0          | 0          | 0          | 0          | 0          | 0          | 0          |
> | SP+SG     | 0          | 0          | 0          | 0          | 0          | 0          | 0          |
> | RoMA      | 0.0854     | 0.0023     | 0          | 0.0036     | 0.5030      | 0          | 0.1388     |
> | SP+RoMa   | 0.3738     | 0          | 0          | 0          | 0.6986     | 0          | 0.5966     |
> | DaD+RoMa  | 0.6941     | 0.8000        | 0          | 0.7915     | 0.5465     | 0.7440      | 0.6380      |
>
> Table 1. Cross-Elevation camera poses obtained from different matching methods. Measured in AUC (higher is better).

---

> ### Author Response · Authors · 2025-11-20
>
> 2. **Standardized tasks/splits**
> > Canonical train/val/test splits by building/elevation/season, the authors should define standardized tasks/splits, cross-elevation dependency on transitional sequences may understate the difficulty when such data are unavailable; direct ground↔aerial matching is intentionally disabled as infeasible.
>
> Our paper focuses on proposing standardized tasks: cross-elevation calibration, scene graph optimization, multi-elevation and multi-appearance dense reconstruction and NVS.
>
> In line 336, we described the setup for cross-elevation calibration, with more details, i.e., number of images and visualization shown in Fig.3 in the main manuscript and Fig. B and C in our Supplemental Material. No splits are involved in calibration, as it is optimization based. Note that **we intentionally removed the dependency of transitional sequence for evaluating cross-elevation calibration**, as visualized in Fig. 3 and Fig. B. For dense reconstruction and NVS, we described our training and test splits in line 375. All setups are the same for every evaluated methods; we will include all splits used in our experiments in the eventual public release. We use default hyperparameters from evaluated methods, with compute budget restrained to 24GB GPUs. If the reviewers have specific requests for reproducibility, we are happy to elaborate with more focus!
>
> 3. Novelty and related work
> >  A semi-automated calibration pipeline with several heuristic/manual steps, which limits both novelty and technical depth. Dataset modest relative to recent city-scale datasets such as Mapillary Metropolis, StreetGaussians, DL3DV-10K, or 4D4NeRF; the paper offers little direct comparison or clear positioning against them.
>
> We note that our submission is under ICLR's Primary Area of **Datasets and Benchmarks**, hence our goal in this work is to introduce a novel dataset and benchmark that explore the lack of a standardized dataset for challenging scenarios for 3D vision. The main body of work explains the validity of the data collection pipeline (such that this process may be replicable to researchers in the future), and the problems revealed through our benchmarking.
>
> However, **there are technical improvements** in this paper outside of the dataset and benchmark content to demonstrate the utility of the ULTRA dataset. As shown in Fig. E in our Supplemental Material, we revealed the spatial-appearance inaccuracy in per-image appearance embedding, and introduced an improvement on 3DGS in-the-wild by leveraging a temporal embedding, which performs just as well as per-image appearance but generalize to different poses. This is only made possible by our dataset, which has multi-view images at different time stamp, unlike Photo Tourism.
>
> We also integrated a sky segmentation model to help with floaters in NVS. Pertinent to another question posed by the reviewer, ULTRA is a strong benchmark for evaluating floaters. Floaters are hard to pinpoint, as geometry is always unknown in inverse rendering (and impossible to obtain groundtruth at scale in such an unconstrained scenario). However, ULTRA datasets contain views that are far from each other, i.e., ground and aerial views. As such, floaters that arise from ground reconstruction can be seen easily from the air, as shown in Fig. 6 in the main manuscript. This is similar to Nerfbusters (2304.10532), which has a much smaller set of data with smaller scene scale.
>
> ULTRA-360 spans ~140 acres with substantial elevation variation, from ground level up to 120 m, and is captured densely so that each region of this volume is observed from multiple angles and elevations. To the best of our knowledge, StreetGaussians evaluates on KITTI-360 and the Waymo Block-NeRF datasets **without proposing new data**, and we could not find an existing dataset called *4D4NeRF*. DL3DV-10K, while a very large-scale collection (10,510 videos, 51.3M frames at 4K across 65 POI categories), is not truly “city-scale”, as it consists of small scenes with small camera variations. Compared to car-captured city-scale datasets such as Mapillary Metropolis (27.7k images), KITTI-360, and Waymo Block-NeRF, ULTRA-360 provides more complete coverage of building elevation and is not constrained to a driving route. Although driving datasets contain more frames, their camera calibration problem is substantially easier. Ultimately, none of these datasets fully demonstrate the challenges in cross-elevation calibration, visual doppelgangers, and multi-elevation multi-appearance NVS. As such, it is a fairly straightforward argument that ULTRA-360 is a novel dataset and benchmark for community to explore and research on.

---

> ### Author Response · Authors · 2025-11-20
>
> 4. Quantitative pose errors against a reference
> >  Can you report quantitative pose errors (e.g., ATE/RE) against a reference (e.g., campus-wide aerial bundle) and per-building success rates for different scene-graph/matcher settings?
>
> Please see Table 2 here for the quantitative pose errors against the campus-wide aerial bundle. To better understand the pose errors, we express it in metric scale (meters). All buildings have centimeter-level accuracy, demonstrating our pipeline’s reliability.
>
> | Building | ATE (m)  | RE (degree) |
> |----------|----------|-------------|
> | #10      | 0.012 | 0.3994      |
> | #24      | 0.008| 0.2689      |
> | #28      | 0.008 | 1.4560       |
> | #30      | 0.004 | 0.0982      |
> | #34      | 0.008 | 0.3291      |
> | #35      | 0.008 | 0.1287      |
> | #48      | 0.008| 0.2925      |
> | #49      | 0.012 | 0.3573      |
> | #53      | 0.004 | 0.0629      |
> | #54      | 0.008 | 0.1185      |
> | #77      | 0.004  | 0.0800      |
> | #78      | 0.008 | 0.2512      |
>
> Table 2. Alignment error, i.e. ATE in meters and rotation error in degree, of individual buildings to campus-wide aerial calibration.
>
> Unfortunately, it is infeasible to iterate all settings of matchers/scene-graphs for such a large-scale, campus-wide calibration of ~40k images, especially with transformer-based matchers. Both the time and space complexity are prohibitive. We refer the reviewer to Table 1 here and Fig. 3 in the main manuscript to demonstrate that RoMa-based matching is the current SoTA in terms of sensitivity. We welcome discussion if the reviewer sees a different conclusion.
>
> 5. GPS to reduce manual labor
> > How sensitive is performance to the front/back bucketing and the sequential window size |i−j|≤10? Could a learned side-classifier (or GPS/compass when available) reduce manual curation?
>
> Front/back bucketing is fairly low effort and robust. Realistically, only two frames per video need to be labeled. From a single sequence perspective, manual curation is potentially not needed with sequential matching; however, multi-sequence/multi-time calibration is where the visual ambiguities are difficult to eliminate, as frames between sequences are not temporally or geometrically related. While things like DP++ (which is similar to a “side detector”), GPS/compass all potentially can help, they are not 100% reliable at scale. Consumer-grade GPS typically has a 2–4-meter accuracy, and DP++ can still lead to doppelgangers as shown in Fig. 4 in the main manuscript. Based on our conclusive manual inspection on all calibration, we do not see any failure cases with front/back bucketing and sequential matching – if the feature matcher does its job.
>
> 6. Validity of doppelganger challenge in relationship with contemporary matchers
> > Could you add/compare LightGlue/MASt3R-SfM and report robustness vs. false positives in ULTRA-360’s doppelgänger regimes?
>
> We reuse the setting from Fig. 4 of the main manuscript: Building #54, which has visually similar front and back doors and a total of 337 images. We evaluate SuperPoint+LightGlue (SP+LG), MASt3R, MASt3R-SfM, VGGT, and VGGT-SfM on this scene to assess their robustness in the doppelganger regime, and report AUC@30 and AUC@10. As shown in Table 3 here, apart from MASt3R, which runs out of memory, most methods fail to reliably distinguish the front door from the back door. SP+SG with DP++ performs well here, but not perfectly, as shown in Fig. C in Supplemental Material. Furthermore, SP+SG fails in cross-elevation scenarios. Again, our dataset reveals the tradeoff between sensitivity and specificity in unconstrained, real world scenarios.
>
> | Method        | AUC@30 | AUC@10 |
> |--------------|--------|--------|
> | SP+LG           | 0.2699 | 0.2593 |
> | MASt3R       | OOM    | OOM    |
> | MASt3R-SfM   | 0.2308 | 0.1706 |
> | VGGT         | 0.1936 | 0.1327 |
> | VGGSfM       | 0.1555 | 0.1291 |
> | DaD+RoMa with DP++  | 0.4897 | 0.476  |
> | SP+SG with DP++     | 0.9583 | 0.9341 |
> | SP+SG with NetVLAD | 0.4967 | 0.482  |
>
> Table 3. Camera pose on Building #54, which reveals doppelganger regime, obtained from different matching methods. Measured in AUC (higher is better).

---

> ### Author Response · Authors · 2025-11-20
>
> 7. Cross-elevation training reduces Gaussian counts
> > Can you isolate whether this is due to optimization schedules, regularization, or scene-graph/pose noise? An ablation on densification thresholds and background/sky modeling across methods would clarify attribution.
>
> We proposed some hypothesis that could cause the cross-elevation densification issue in line 433 of the manuscript. This is a problem that have been similarly observed by others other scenarios, e.g. Revising Densification in Gaussian Splatting (Bulò et al.) and Taming 3DGS (Mallick et al.). Pose noise is unlikely to be the major cause, as we have the same observations in synthetic scenes where the pose is noise free. Background/sky modeling is also unlikely, as the sky can only be seen by ground images and does not affect the dynamics of ground-aerial joint optimization.
>
> We note that in Table 4 of the main manuscript, all experiments are done with the same densification threshold, i.e. the same densification threshold worked well for ground and aerial images individually, but not together. A meaningfully different densification threshold will hurt the individual performances, thus not solving the problem. As an ongoing research direction, we are investigating the concept of sequential Gaussian optimization, i.e., fixing ground Gaussians after ground-only optimization, then perform aerial-only optimization by introducing newer Gaussians. That resolves the Gaussian reduction issue and leads to better quality, but is out of the scope for this work.
>
> 8. Privacy concerns
> > Campus imagery may include bystanders and vehicles. You blur PII (faces, plates), which is good; still, please clarify detector specifics, thresholds, manual QA, and policies for takedown requests. Also clarify whether minors might appear and how you handle that.
>
> We design our collection process to respect privacy, especially during peak campus activity: as a rule we avoid data acquisition when large crowds are present and focus on building façades rather than people. For PII removal, we run automatic detection and blurring using InsightFace for faces, a YOLO-VOC–style detector for vehicles, and WPODNet for license plates, all with a detection confidence threshold of 0.5. After automated processing, we manually review all images to verify that faces and license plates are fully removed or blurred; images with residual PII are either corrected or discarded. Our collection is restricted to university campus areas and we intentionally avoid minors and discard frames with minors in them; in any case, all detected faces are treated identically and blurred.

---

> ### Author Response · Authors · 2025-11-26
>
> Dear Reviewer S12u,
>
> We kindly invite you to review our rebuttal since the discussion phase is ending soon. We have carefully addressed all your concerns, including:
>
> 1.	Validity of the cross-view and doppelganger calibration challenges presented by this dataset, especially in relationship to contemporary feed-forward matchers.
> 2.	Novelty and reproducibility
> 3.	Reliability of our registration pipeline
> 4.	Manual labor usage
> 5.	Explanation on gaussian count reduction
> 6.	Privacy concerns
>
> If you have any further questions or would like additional clarifications, we would be happy to continue the discussion.

---

### Meta-Review · Area_Chair_DEyq · 2026-01-06

**Summary:**

The paper introduces ULTRA-360, a large-scale, multi-modal dataset designed to benchmark 3D and 4D scene reconstruction in unconstrained environments. The dataset captures a university campus (~140 acres) over two years, featuring imagery across four seasons and various weather conditions. A key distinguishing feature is its integration of diverse viewpoints, combining ground-level omnidirectional (360°) imagery with aerial drone footage at multiple elevations (60m-120m).

To establish accurate camera poses, the authors propose a semi-automated calibration pipeline using a divide-and-conquer strategy. This includes manual verification to resolve "doppelgangers" visual ambiguities (repetitive architectural features) that typically confuse automated Structure-from-Motion (SfM) pipelines. The paper benchmarks various SOTA methods for feature matching and dense reconstruction (e.g., 3DGS variants), revealing significant limitations in current methodologies regarding cross-elevation matching and densification issues when bridging ground and aerial perspectives.

Though the novelty and initial experiments are limited, I recommend accepting this paper **if there is a "Datasets and Benchmarks" trend under ICLR's Primary Area**. ULTRA-360 fills a genuine gap by bridging the ground-to-air domain with temporal variations. While Reviewer o2rg notes the calibration is an "engineering effort," compiling high-quality Ground Truth for large-scale outdoor scenes is a significant contribution that enables future research. The authors' rebuttal was exceptionally strong: they empirically proved that modern baselines (MASt3R, DUSt3R) fail on this data, demonstrating that the dataset poses a necessary challenge to the community.

**Reviewer Concerns:**

The three reliable reviewers (o2rg, eWid, oFTk) appreciate the dataset’s scale, diversity, and calibration effort. Common concerns include: (1) limited methodological novelty beyond data collection; (2) initial omission of some contemporary matchers; (3) need for clearer evaluation splits and reproducibility details; (4) reliance on manual steps in the pipeline.

**Reviewer Scores:**

The scoring spread reflects a mix of positive assessments and concerns about novelty.

*   **Reviewer eWid:** Score 6 (Marginally Above Acceptance) - Appreciated the motivation and scale.
*   **Reviewer oFTk:** Score 6 (Marginally Above Acceptance) - Valued the 4D aspects.
*   **Reviewer o2rg:** Score 4 (Marginally Below Acceptance) - Concerned about novelty (though the rebuttal addressed baseline requests).

---

### Decision · Program_Chairs · 2026-01-26

Accept (Poster)